



# Partitioning of hydrogen peroxide in gas-liquid and gas-aerosol phases

Xiaoning Xuan[1], Zhongming Chen[1], Yiwei Gong[1], Hengqing Shen[1], and Shiyi Chen[1]

[1]State Key Laboratory of Environmental Simulation and Pollution Control, College of Environmental Sciences and Engineering, Peking University, Beijing, 100871, China

*Correspondence to*: Zhongming Chen (zmchen@pku.edu.cn)

**Abstract.** Hydrogen peroxide ($H_2O_2$) is a vital oxidant in the atmosphere and plays critical roles in the oxidation chemistry of both liquid and aerosol phases. The partitioning of $H_2O_2$ between the gas and liquid phase or the aerosol phase could affect its abundance in these condensed phases and eventually the formation of secondary components. However, the partitioning processes of $H_2O_2$ in gas-liquid and gas-aerosol phases are still unclear, especially in the ambient atmosphere. In this study, field observations of gas-, liquid-, and aerosol-phase $H_2O_2$ were carried out in the urban atmosphere of Beijing during the summer and winter of 2018. The effective field-derived mean value of Henry's law constant ($H_A^m$, $2.1 \times 10^5$ M atm$^{-1}$) was 2.5 times that of the theoretical value in pure water ($H_A^t$, $8.4 \times 10^4$ M atm$^{-1}$) at $298 \pm 2$ K. The effective derived gas-aerosol partitioning coefficient ($K_P^m$, $3.8 \times 10^{-3}$ m$^3$ µg$^{-1}$) was four orders of magnitude higher on average than the theoretical value ($K_P^t$, $2.8 \times 10^{-7}$ m$^3$ µg$^{-1}$) at $270 \pm 4$ K. The partitioning of $H_2O_2$ in the gas-liquid and gas-aerosol phases in the ambient atmosphere does not only obey Henry's law or Pankow's absorptive partitioning theory but is also influenced by certain physical and chemical reactions. The average concentration of liquid-phase $H_2O_2$ in rainwater during summer was $44.12 \pm 26.49$ µM. In three-quarters of the collected rain samples, the measured $H_2O_2$ was greater than the predicted value in pure water calculated by Henry's law. In these samples, 46 % of the measured $H_2O_2$ was from gas-phase partitioning, and most of the rest may have come from residual $H_2O_2$ in raindrops. In winter, the level of aerosol-phase $H_2O_2$ was $0.093 \pm 0.085$ ng µg$^{-1}$, which was much higher than the predicted value based on Pankow's absorptive partitioning theory. Almost all aerosol-phase $H_2O_2$ was not from the partitioning of the gas phase. The decomposition/hydrolysis of aerosol-phase organic peroxides could be responsible for 32 % of aerosol-phase $H_2O_2$ formation at the maximum rate of 3.65 ng µg$^{-1}$. Furthermore, the heterogeneous uptake of $H_2O_2$ on aerosols contributed to less than 0.5 %.

## 1 Introduction

Hydrogen peroxide ($H_2O_2$), regarded as a significant oxidant in the liquid and aerosol phases, is of great significance to the oxidation capacity in these phases (Reeves and Penkett, 2003). In addition to its direct effect as an oxidant, $H_2O_2$ also serves as the temporary reservoir species that cycles and redistributes the $HO_x$ radical (Lee et al., 2000; Tong et al., 2016; Crowley et al., 2018). Owing to larger solubility in water (O'Sullivan et al., 1996) and larger reaction rate with reduced substances



(Seinfeld and Pandis, 2006), $H_2O_2$ plays a vital part in the fast formation of sulfate ($SO_4^{2-}$) and fine particles ($PM_{2.5}$) during heavy haze episodes (Stein and Saylor, 2012; Qin et al., 2018; Ye et al., 2018; Liu et al., 2020). Furthermore, $H_2O_2$, as a typical reactive oxygen species (ROS), has adverse health effects and contributes to incidences of lung cancer, asthma, and cardiopulmonary disease (Gurgueira et al., 2002; Zhao et al., 2011; Campbell et al., 2019).

As is well known, $H_2O_2$ in the liquid and aerosol phases is generally assumed to originate from the partitioning of gas-phase
$H_2O_2$. Furthermore, the partitioning of $H_2O_2$ between the gas-liquid and gas-aerosol phases is expected to obey Henry's law and Pankow's absorptive partitioning theory, respectively. Studies of the partitioning process of $H_2O_2$ contribute to a clearer understanding of the sources of limiting oxidants and estimates of the contribution to sulfate formation in the liquid and aerosol phases. In this study, we define the field-derived ratios of the measured levels of gas-to-liquid and gas-to-aerosol phases as the effective Henry's law constant and the gas-aerosol partitioning coefficient, respectively.

However, it is interesting that the predicted liquid-phase concentration of $H_2O_2$ in rainwater using Henry's law was not sufficient to account for the measured level, and a large amount of liquid-phase $H_2O_2$ was produced from other reactions (Liang et al., 2013). Chung et al. (2005) demonstrated that the "salting-in" effect could as much as double the solubility of $H_2O_2$ in salt solutions up to 10 M. However, the ionic strength in rainwater is too low to impose the "salting-in" effect (Li et al., 2019). Therefore, we need to seek other explanations. In addition, the gas-phase $H_2O_2$ level at the ground after a shower was higher
than before the shower, revealing that raindrops could release $H_2O_2$ into the gas phase at the ground (Hua et al., 2008). This opens up new possibilities for explaining the high level of $H_2O_2$ in rainwater. Nevertheless, the falling of raindrops is a complex process that involves several uncertainties, so observation studies are needed to quantitatively explain the high concentration in rainwater.

The measured level of $H_2O_2$ in aerosol particles was much higher than the theoretical value for gas-aerosol partitioning, by
two orders of magnitude (Hasson and Paulson, 2003; Arellanes et al. 2006). In previous studies, it was confirmed that considerable $H_2O_2$ could be produced from redox reactions in aerosols, like transition metals (Charrier et al., 2014). However, it is noticeable that continuous redox reactions are assisted by available reductants, so it is impossible for ambient aerosols to generate $H_2O_2$ from transition metals without an additional reduced agent (Shen et al., 2011). Recently, numerous studies have reported the decomposition of organic peroxides in the aerosol phase (Krapf et al., 2016; Riva et al., 2017). Li et al. (2016)
suggested that the decomposition/hydrolysis of organic peroxides on secondary organic aerosol particles could substantially raise the level of $H_2O_2$. Qiu et al. (2019) proposed that α-hydroxyalkyl-hydroperoxides could be easily decomposed into $H_2O_2$ within 2 h in ≥ 10 % water mixtures. However, the quantitative measurement of organic peroxides is difficult because of their instability (Zhao et al., 2018). The decomposition of labile organic peroxides should be studied in atmospheric fine particles ($PM_{2.5}$) in heavily polluted areas. In addition, $H_2O_2$ is easily to adsorbed onto aerosol particles, its heterogeneous uptake should
also be considered. Hence, a quantitative evaluation of sources other than gas-phase partitioning is needed with the support of field measurements.

Compared to gas-phase $H_2O_2$, it is challenging to quantitatively understand the chemistry of $H_2O_2$ in the liquid and aerosol phases. To the best of our knowledge, this is the first study to measure $H_2O_2$ in gas-liquid or gas-aerosol phases simultaneously



in a heavily polluted area, e.g., Beijing, providing a good opportunity to better understand the partitioning of $H_2O_2$ in different
phases. The objectives of this study are to explore the partitioning of $H_2O_2$ in the gas-liquid and gas-aerosol phases in the
ambient atmosphere and to seek possible sources other than gas-phase partitioning that could increase $H_2O_2$ concentration in
the liquid and aerosol phases.

## 2 Experimental

### 2.1 Measurement site

The online gas-phase measurement of peroxides was performed at the Peking University (PKU) site (39.99° N 116.30° E),
situated in the northwest of urban Beijing. The PKU site is a typical city site in a heavily polluted area in Beijing, with two
main trunks of traffic to the east and south. The relative apparatuses were placed on the roof of a building that was ~ 26 m
above ground level. In this study, we introduce two measurements at the PKU site: BJ-2018Summer (23 July−10 August 2018
and 25 August−11 September 2018) and BJ-2018Winter (21 December 2018−5 January 2019).

### 2.2 Measurement methods

### 2.2.1 Gas-phase peroxides

The concentrations of gas-phase peroxides were observed in both BJ-2018Summer and BJ-2018Winter using high performance
liquid chromatography (HPLC, Agilent 1200, USA) with a time resolution of 21 min. The HPLC coupled with the post-column
enzyme derivatization method could distinguish $H_2O_2$ from organic peroxides. This method is well established (Hua et al.,
2008; He et al., 2010) and is only briefly described here. Ambient air was drawn into a glassy scrubbing coil at a flow rate of
2.7 standard L min$^{-1}$. $H_3PO_4$ solution ($5 \times 10^{-3}$ M) was added to the scrubbing coil at 0.2 mL min$^{-1}$ to dissolve $H_2O_2$ from
ambient air. The collection efficiency of $H_2O_2$ was validated to close to 100 %, while it was ~ 85 % for organic peroxides.
Then, the mixture was injected into HPLC with the mobile phase ($H_3PO_4$, $5 \times 10^{-3}$ M). Peroxides separated by the column
reacted stoichiometrically with parahydroxyphenylacetic acid (PHPAA) under the Hemin catalyst, generating stable PHPAA
dimers that were measured by a fluorescence detector. The peroxides were identified and quantified using standard samples,
and the detection limit (DL) of the gas-phase $H_2O_2$ was about 10 pptv. The values below DL were replaced by DL divided by
the square root of two (the same hereafter). The gas-phase samples during BJ-2018Summer were used for the partitioning
analysis in the gas-liquid phase, while the data of BJ-2018Winter were used to study the partitioning in the gas-aerosol phase.

### 2.2.2 Liquid-phase peroxides

Rain samples were collected by a custom-built glass funnel and used for the analysis of liquid-phase peroxides in BJ-
2018Summer. During the observation period, the collection of rain samples was well organized depending on the intensity,
amount, and duration of the rain. Because the peroxides were easy to break down, the collected rain samples were preserved



in brown vials at 4 °C until they could be analysed with HPLC within 6 h. The subsequent detection method for the liquid-phase peroxides was the same as for the gas-phase peroxides. In all, we collected 60 rain samples during seven rain episodes, and the DL of the liquid-phase $H_2O_2$ was about 8 nM. The specific dates of the rain events in chronological order were 24 July, 25 July, 5 August, 6 August, 8 August, 30 August and 2 September.

### 2.2.3 Aerosol-phase peroxides

Aerosol-phase samples were gathered on Teflon filters (Whatman™, 47 mm diameter and 2 μm pore size) using a four-channel filter sampler (Wuhan Tianhong TH-16A, China) at 16.7 standard L min$^{-1}$ during BJ-2018Winter. Teflon filters were supported by stainless steel filter holders during the 11.5 h sampling time. We immediately disposed of two Teflon filters for the analysis of peroxides and total peroxides (TPO$_S$), and the remaining filters were kept under refrigeration at −18 °C for subsequent component analysis. For analysing the aerosol-phase peroxides, the Teflon filters were immediately extracted with 10 mL $H_3PO_4$ in conical flasks and placed on a shaker to be blended thoroughly at 4 °C and 180 rpm for 15 min. Then, the extracted solution was measured with HPLC within 40 min. The extracted solution was also used for the measurement of TPOs using the iodometric spectrophotometric method, which could measure $H_2O_2$ as well as organic peroxides. (Nozaki, 1946; Banerjee and Budke, 1964). After the oxygen in the extracted solution was blown off by bubbling with nitrogen for 5 min, 250 μL potassium iodide solution (KI, 0.75 M) was added to the solution to react with TPOs in the dark for 12−24 h (Reactions R1 and R2). The reaction product $I_3^-$ ion could be detected using UV/Vis spectrophotometry (Beijing PERSEE TU-1810, China) at the wavelength of 420 nm. A total of 31 aerosol-phase samples were analysed, and the DL of aerosol-phase $H_2O_2$ was close to 0.24 ng m$^{-3}$ (0.006 ng μg$^{-1}$).

To avoid the matrix influence on samples (i.e., Teflon filters and the $H_3PO_4$ solution), we measured the concentration of blank samples in every extraction. The level of $H_2O_2$ in three-quarters of the blank samples was equal to 0 μM, and the concentration of $H_2O_2$ in the remnant blank samples was below 10 % of that in the ambient air samples. To prevent the matrix influence, we deducted the background values of the samples. In addition, to ensure that the measured $H_2O_2$ was attributed to aerosols collected on Teflon filters, we performed experiments to demonstrate that the physical adsorption on clean Teflon filters without aerosols was responsible for 15 % of the measured $H_2O_2$ in samples. The details are available in Fig. S1 in the Supplement. In this study, we did not correct the physical adsorption to avoid introducing new errors.

$$3I^- + H_2O_2 + 2H^+ \rightarrow I_3^- + 2H_2O \tag{R1}$$

$$R_1OOR_2 + 3I^- + 2H^+ \rightarrow I_3^- + R_1OH + R_2OH \tag{R2}$$

### 2.2.4 Other components and meteorological parameters

Water-soluble cations (Na$^+$, NH$_4^+$, K$^+$, Mg$^{2+}$, and Ca$^{2+}$) as well as anions (Cl$^-$, NO$_3^-$, and SO$_4^{2-}$) were measured with ion chromatography (IC, Dionex ICS2000 and ICS2500, USA). Transition metal elements deposited on Teflon filters were measured with inductively coupled plasma mass spectrometry (ICP-MS, Bruker aurora M90, Germany). The mass



concentration of $PM_{2.5}$ was measured with a TEOM 1400a analyser. Meteorological parameters (temperature, relative humidity, and wind speed) and major trace gases ($O_3$, $SO_2$, $NO$-$NO_2$-$NO_x$ and $CO$) were monitored simultaneously using a series of commercial instruments (Met One Instruments Inc., Thermo 49i, 43i, 42i, and 48i).

## 2.3 Estimation of effective partitioning coefficients

To estimate the effective partitioning coefficients, we could use the field-derived Henry's law constant for the gas-liquid phase and the gas-aerosol partitioning coefficient for the gas-aerosol phase (Pankow, 1994), which are estimated according to Eqs. (1)–(4).

$$H_A^t = 8.4 \times 10^4 \ M \ atm^{-1} \tag{1}$$

$$H_A^m = \frac{C_{aq}^m}{C_g^m} \tag{2}$$

$$K_P^t = \frac{RT_W f_{om}}{10^6 \overline{MW_{OM}} \zeta p_L^0} \tag{3}$$

$$K_P^m = \frac{C_p^m}{C_g^m C_{om}} \tag{4}$$

In Eqs. (1) and (2), $C_{aq}^m$ is the liquid-phase level of $H_2O_2$, M; $C_g^m$ is the partial pressure of the gas-phase $H_2O_2$, atm; and $H_A^t$ and $H_A^m$ are the theoretical values in pure water and the effective field-derived Henry's law constant, respectively, M atm$^{-1}$ (Sander et al., 2011). The average temperature during rainfall in summer ($T_S$) was 298 ± 2 K (mean ± standard deviation, the same hereafter). In Eq. (3), $K_P^t$ is the theoretical value of the gas-aerosol partitioning coefficient, m$^3$ μg$^{-1}$; $\overline{MW_{OM}}$ is the estimated average molecular weight of organic compounds, 200 g mol$^{-1}$ (Williams et al., 2010; Xie et al., 2014); $p_L^0$ is the vapour pressure of pure $H_2O_2$ at the specified temperature, calculated by the extrapolation of the Antoine equation (Maass and Hiebert, 1924; Baum et al., 1997); $\zeta$ is the activity coefficient of $H_2O_2$, assumed to be unity (Pankow, 1994); $f_{om}$ is the weight fraction of $H_2O_2$ absorbing on aerosols, also set to unity (Liang et al., 1997; Shen et al., 2018); $R$ is the ideal gas constant, 8.2 × 10$^{-5}$ atm m$^3$ mol$^{-1}$ K$^{-1}$; and $T_W$ is the mean temperature during BJ-2018Winter, 270 ± 4 K for the whole observation period, 272 ± 4 K for day-time, and 269 ± 4 K for night-time. In Eq. (4), $C_p^m$ and $C_g^m$ are the concentrations of $H_2O_2$ in the aerosol and gas phases, respectively, μg m$^{-3}$; $C_{om}$ is the organic matter concentration, referring to the mass concentration of $PM_{2.5}$, μg m$^{-3}$; and $K_P^m$ is the effective field-derived gas-aerosol partitioning coefficient, m$^3$ μg$^{-1}$.

## 3 Results and discussion

### 3.1 Gas-liquid phase partitioning

#### 3.1.1 Gas- and liquid-phase $H_2O_2$ in summer

The concentration of gas-phase $H_2O_2$ was statistically counted to be 0.30 ± 0.26 parts per billion by volume (ppbv) for the



seven rainfalls (Fig. S2a in the Supplement) and $0.53 \pm 0.77$ ppbv for the entire BJ-2018Summer. Compared with the theoretical

liquid-phase $H_2O_2$ value in pure water with 25.20 μM, the level of measured $H_2O_2$ in the liquid phase was $44.12 \pm 26.49$ μM

(3.19−139.95 μM), as shown in Fig. S2b. The detailed values of the peroxides in the gas and liquid phases are shown in Table

S1. Based on Eq. (2), the effective field-derived Henry's law constant, $H_A^m$, averaged $2.1 \times 10^5$ M atm$^{-1}$ in rain samples, which

was two and a half times the theoretical pure-water Henry's law constant, $H_A^t$, at $8.4 \times 10^4$ M atm$^{-1}$ and $298 \pm 2$ K. The analysis

result shows that 88 % of the measured liquid-phase $H_2O_2$ came from gas-phase partitioning, while 12 % of $H_2O_2$ was from

other sources. In 23 % of the total rain samples, $H_A^m$ was less than $H_A^t$, indicating that these samples followed Henry's law

(Fig. 1). In the remaining 77 % of the samples, the measured liquid-phase $H_2O_2$ was larger than the predicted values (Fig. 1),

and the difference averaged 28 μM with a maximum of 71 μM. Further, 54 % of liquid-phase $H_2O_2$ in these samples was

produced from other sources than gas-phase partitioning.

To explain the difference between $H_A^m$ and $H_A^t$, we should rule out the effect of pressure, pH, and $T_S$ on $H_A^t$. First, to our

knowledge, the influence of pressure on $H_A^t$ can usually be neglected under conditions of less than 1 atm (Lind and Kok,

1986). Also, $H_A^t$ of $H_2O_2$ is independent of pH in the range of 4−7 (Xu et al., 2012); therefore, the present study does not

consider the influence of pressure and pH on $H_A^t$. The temperature during BJ-2018Summer can be divided into three ranges:

294−296 K, 297−299 K, and 300−306 K. The percentage of samples in these three temperature ranges were 25 %, 63 %, and

12 %, respectively, and the ratios of $H_A^m$ to $H_A^t$ in the same temperature ranges were 1.4, 2.6, and 4.5, respectively. The

maximum value of $H_A^t$ in the range 294−306 K was $1.2 \times 10^5$ M atm$^{-1}$, while $H_A^m$ reached $4.2 \times 10^5$ M atm$^{-1}$ at the 90th

percentile. This suggests that the influence of $T_S$ on $H_A^m$ was negligible. The nonlinear relationship between $H_A^m$ and $T_S$,

shown in Fig. S3, also indicates that $T_S$ plays an unimportant role in determining $H_A^m$. Thus, other explanations need to be

explored to understand the difference between $H_A^m$ and $H_A^t$.

## 3.1.2 Process of raindrops falling

The solubility of $H_2O_2$ in clouds was larger than that in the ground rainwater. There is a negative dependence of the solubility

on temperature (Huang and Chen, 2010), which allows for the possibility of mass transfer of $H_2O_2$ from rainwater to the

surrounding air when falling. Let us assume that the gas-phase $H_2O_2$ concentration is homogeneous and the rain droplet size

remains constant during its fall. The diameter of the raindrops ($D_P$) is mainly distributed in the range of 0.05–2.50 mm.

Calculations were performed for typical droplet diameters at 0.1 mm, 0.5 mm, 1.0 mm and 2.0 mm. The height of the

precipitation cloud base during summer time in north China was almost less than 2000 m (Shang et al., 2012). As a result, we

assumed the fall distance to be 500 m, 1000 m, 1500 m, and 2000 m, respectively, which are same to previous studies

(Adamowicz, 1979; Levine and Schwartz, 1982). In the process of falling, it is necessary to consider the mass transfer

resistance in the gas and liquid phases. However, it could be that the shear force generated on the raindrop surface when it fell

improved the mixing rate in the droplet significantly; therefore, the liquid-phase mass transfer resistance was negligible

(Pruppacher and Klett, 1997; Elperin and Fominykh, 2005). Thus, the overall mass transfer resistance reduced to the mass



transfer resistance in the gas phase.

Here, we first discuss residual $H_2O_2$ in raindrops after a fall from a height of 1000 m. The temperature in clouds ($T_S^c$) was estimated to be 292 K, 6 K lower than the ground. $H_A^t$ in pure water at 292 K is $1.4 \times 10^5$ M atm$^{-1}$ (Sander et al., 2011). Provided that the droplet started at equilibrium with the cloud atmosphere, the initial level of liquid-phase $H_2O_2$ before falling ($C_{aq}^0$) was 42.87 μM. However, the equilibrium was broken when the raindrops fell as the ambient temperature increased. The mass transfer coefficient in the gas phase ($k_g$) can be calculated by Eqs. (S1)−(S4) in the Supplement (Levine and Schwartz, 1982; Kumar, 1985). The concentration of $H_2O_2$ in the droplet at the ground ($C_{aq}^d$) can be estimated by Eq. (S5). The results are presented in Table 1, which shows us that the large droplet has a small mass transfer coefficient. As a result, the liquid-phase $H_2O_2$ in the large raindrops is more slowly released into the air. $C_{aq}^d$ of the droplet diameter at 2.0 mm was close to $C_{aq}^0$, while $C_{aq}^d$ at 0.1 mm approximated the theoretical level of liquid-phase $H_2O_2$ in pure water at 298 K, as indicated by Fig. 2. The results show that the effect of residual $H_2O_2$ in large raindrops on ground rainwater levels could be of great importance. Next, we investigated the influence of different fall distances on $C_{aq}^d$. The decreasing temperature at increasing fall distances caused larger $H_A^t$ and $C_{aq}^0$ in clouds, and $C_{aq}^d$ also increased. The wide gap of $C_{aq}^d$ between different fall distances is more visible for the large droplet, as seen in Fig. 2. Based on the above analysis, the residual $H_2O_2$ in large raindrops can increase the $H_2O_2$ level in rainwater to a maximum of 48.81 μM at a fall distance of 2000 m, which explains to a large extent the difference between the measured and predicted levels of $H_2O_2$ in rainwater.

Based on the rain intensity, seven rain events during BJ-2018 Summer could be divided into three types, as shown in Table S2 in the Supplement. Rain events in types I, II, and III have rain intensities < 1 mm h$^{-1}$, 1−10 mm h$^{-1}$, and > 10 mm h$^{-1}$, respectively. The larger the diameter of raindrops, the greater the rain intensity (Kumar., 1985). According to the above relationship between the diameter of raindrops and level of liquid-phase $H_2O_2$ in the ground rainwater, the difference between the measured and predicted liquid-phase $H_2O_2$ levels should be greater as the hourly rain intensity increased. We found that the differences between $C_{aq}^m$ and $C_{aq}^t$ increased during the rain periods on 25 July and 5 August, which the maximum hourly rain intensities were more than 10 mm h$^{-1}$. Because it is difficult for the liquid-phase $H_2O_2$ in heavy rains to diffuse into the gas phase, much $H_2O_2$ may be retained in the ground rainwater, which could well represent the level of $H_2O_2$ in cloud water. During the rain episode on 1−2 September 2018, the concentration of gas-phase $H_2O_2$ decreased over time. However, there was a sudden rise from 0.47 ppbv at 1:03 local time (LT) to 0.66 ppbv at 1:46 LT, which subsequently dropped to 0.38 ppbv over time (Fig. 3a). Surprisingly, the difference between the measured and predicted levels of liquid-phase $H_2O_2$ reached a low value in the meantime, indicating that the increase in gas-phase $H_2O_2$ was due to the release of $H_2O_2$ from raindrops that could contain high levels of $H_2O_2$, as presented in Fig. 3b. Compared with Fig. S4 in the Supplement, which describes the relationship between rain intensity and time, the rain intensity simultaneously dropped to 3.51 mm h$^{-1}$ from 6.35 mm h$^{-1}$, consequently decreasing the diameter of the raindrops (Kumar, 1985) and increasing the mass transfer of $H_2O_2$ from rainwater to the gas phase. Provided that 20 μM $H_2O_2$ in rainwater was released into ambient air, the increase in the gas-phase $H_2O_2$ level was 0.24 ppbv, which was in accordance with the sudden rise during 1:03−1:46 LT on 1−2 September 2018.



The above analysis is based on the assumption that the gas-phase $H_2O_2$ concentration is uniform. However, the distribution of gas-phase $H_2O_2$ at different heights may be complicated. We could use the average level of $H_2O_2$ in rainwater at the ground to estimate the concentrations of $H_2O_2$ in cloud water ($C_{aq}^c$) and the nearby atmosphere ($C_g^c$), as presented in Table 2. Assuming the simplest case, $D_P$ is 1.0 mm, the fall distance is 1000 m, and $H_2O_2$ in the gas phase and rainwater at the ground is 0.30 ppbv and 44.12 μM at 298 K. Considering the release of $H_2O_2$ from raindrops into ambient air during the falling process, the

level of $H_2O_2$ in cloud water should be 47 μM. Based on Henry's law, the surrounding gas-phase $H_2O_2$ may be 0.33 ppbv, a little higher than that at the ground. When the fall distance is 500 m, 1500 m, and 2000 m, $H_2O_2$ in cloud water should be 46 μM, 49 μM, and 51 μM, respectively, and $H_2O_2$ in nearby ambient air could be 0.41 ppbv, 0.26 ppbv, and 0.21 ppbv, respectively.

### 3.2 Gas-aerosol phase partitioning

#### 3.2.1 Gas- and aerosol-phase $H_2O_2$ in winter

From 21 December 2018 to 5 January 2019, the gas-phase $H_2O_2$ level was $24.08 \pm 28.83$ parts per trillion by volume (pptv), as shown in Fig. S5a in the Supplement. We eluted Teflon filters with $H_3PO_4$ solution and measured the level of $H_2O_2$ in the extracted solution to calculate the aerosol-phase $H_2O_2$ concentration. The mass concentration of aerosol-phase $H_2O_2$ and the normalized concentration to aerosol mass were $2.22 \pm 1.49$ ng m$^{-3}$ ($< 0.24-6.75$ ng m$^{-3}$) and $0.093 \pm 0.085$ ng μg$^{-1}$ ($<$

$0.006-0.409$ ng μg$^{-1}$), respectively. The mean concentration of the aerosol-phase $H_2O_2$ at night-time ($0.107 \pm 0.102$ ng μg$^{-1}$) was higher than that at day-time ($0.079 \pm 0.066$ ng μg$^{-1}$), as presented in Fig. S5b. The measured level of $H_2O_2$ in aerosols was much higher than the predicted value using Pankow's absorptive partitioning theory, which suggested that the aerosols collected on the filter existed under non-equilibrium conditions and may arise from sources other than gas-phase partitioning in the aerosol phase. Based on Eqs. (3) and (4), $K_P^m$ was equal to $3.8 \times 10^{-3} \pm 4.8 \times 10^{-3}$ m$^3$ μg$^{-1}$ at $270 \pm 4$ K, which was four

orders of magnitude more than $K_P^t$, $2.8 \times 10^{-7}$ m$^3$ μg$^{-1}$. The effects of parameter variation (e.g., $T_W$, $\zeta$, $\overline{MW_{OM}}$, and $f_{om}$) could not account for the large discrepancy between $K_P^m$ and $K_P^t$ (Shen et al., 2018), and other factors are needed to explain the difference. In terms of the proportion of theoretical to measured concentrations, the partitioning of gas-phase $H_2O_2$ into aerosols could be neglected, and nearly all of aerosol-phase $H_2O_2$ was generated from other reactions besides gas-phase partitioning.

The level of aerosol-phase $H_2O_2$ in the present study was lower than those reported in previous studies (Table S3), which may be due to the extraction method, the extraction time, reduced substance levels, and aerosol pH values, as shown in the Supplement. Assuming a molecular weight of 300 g mol$^{-1}$ (Docherty et al., 2005; Epstein et al., 2014), the level of TPOs averaged $10.26 \pm 6.38$ ng μg$^{-1}$ ($2.08-28.75$ ng μg$^{-1}$). It was calculated that $H_2O_2$ took up a small fraction of TPOs, equal to $8 \pm 6$ % in molar concentration ratio, which indicated that organic peroxides accounted for a large proportion of peroxides, and

could play important roles in the formation of PM$_{2.5}$ and secondary organic aerosols.





### 3.2.2 Factor analysis

Figure 4 shows that the concentration of aerosol-phase $H_2O_2$ is dependent on RH, first increasing and then decreasing as RH increases. The variation of $H_2O_2$ with RH was the result of competition between production and consumption processes. Here, the production process refers to either process that favours increasing the level of aerosol-phase $H_2O_2$, while the consumption process denotes those processes consuming aerosol-phase $H_2O_2$. In the first stage, the higher RH could accelerate the heterogeneous uptake of $H_2O_2$ onto aerosols and enhance the level of aerosol-phase $H_2O_2$ (Pradhan et al., 2010; Shiraiwa et al., 2011; Zhao et al., 2013; Slade and Knopf, 2015). The level of $H_2O_2$ is negatively associated with RH in the subsequent stage, ascribed to the much more rapid consumption of $H_2O_2$ due to its oxidizing the reduced substances, such as $SO_2$ into $SO_4^{2-}$, on polluted days.

We considered a heavy haze episode, from 2 January to 3 January 2019, as an example to explain in detail the important contribution of aerosol-phase $H_2O_2$ to $SO_4^{2-}$ growth on polluted days. The $PM_{2.5}$ mass concentration of the severe haze event was up to 201.20 $\mu$g m$^{-3}$. The estimation of aerosol water content is shown in the Supplement. Based on the measured $H_2O_2$, the reaction rate (RR) and sulfate formation rate (SFR) averaged about $3.03 \times 10^{-3}$ $\mu$mol m$^{-3}$ h$^{-1}$ and 0.29 $\mu$g m$^{-3}$ h$^{-1}$ (Table S4), respectively. The detailed calculation process is provided in the Supplement. In addition, the growth rate of $SO_4^{2-}$ calculated by the measured data was 0.51 $\mu$g m$^{-3}$ h$^{-1}$, accounting for about 9 % of the observed formation of $PM_{2.5}$. $H_2O_2$ oxidation pathway contributed about 57 % of the measured growth of $SO_4^{2-}$ in $PM_{2.5}$. This result strongly suggests that the aerosol-phase $H_2O_2$ indeed acts as the important oxidant in the formation of sulfate and plays significant roles in the rapid growth of $PM_{2.5}$ during pollution events.

Next, we consider that the consumption rate of aerosol-phase $H_2O_2$ increases with an increase of RH. It is suggested that larger levels of $SO_4^{2-}$ and $PM_{2.5}$ are often accompanied by higher RH. In Fig. 5, the reverse curve between aerosol-phase $SO_4^{2-}$ and $H_2O_2$ got steeper with the formation of $SO_4^{2-}$, indicating that the rate of consumption of $H_2O_2$ on polluted days was much higher than that on clear days, which could offer proof of rapid $H_2O_2$ consumption with increasing RH. In addition, the level of $H_2O_2$ in the aerosol phase exhibited a negative correlation with $PM_{2.5}$ mass concentration, as shown in Fig. 5. In other words, the aerosol-phase $H_2O_2$ concentration was lower on polluted days than on clear days, which further demonstrates that the removal rate of $H_2O_2$ by oxidizing $SO_2$ into $SO_4^{2-}$ exceeded the production rate during pollution events with high RH.

### 3.2.3 Heterogenous uptake of $H_2O_2$

In addition to the factors that influencing the aerosol-phase $H_2O_2$ concentration, there are other physical and chemical reactions besides gas-phase partitioning that increase the level of aerosol-phase $H_2O_2$, e.g., heterogeneous uptake of $H_2O_2$ on aerosols. Previous studies have shown that heterogeneous uptake of $H_2O_2$ is positively related with RH. High RH is beneficial to the mass transfer of $H_2O_2$ from the gas phase to the aerosol phase, accelerating the reaction between $H_2O_2$ and reduced compositions of aerosols, thus contributing to the more heterogenous uptake of $H_2O_2$ (Huang et al., 2015; Wu et al., 2015). To quantitatively evaluate the importance of the heterogeneous uptake of $H_2O_2$ on aerosols to the aerosol-phase $H_2O_2$, we



calculated the average of total heterogeneous uptake during the sampling process based on Eqs. (S6)−(S11) in the Supplement. Details about each parameter are introduced in Table 3. Compared with the averaged measured content of aerosol-phase $H_2O_2$

($[X]_p^m$, 0.057 ng µg$^{-1}$), the total heterogeneous uptake of $H_2O_2$ ($[X]_p^{t,h}$) averaged 0.049 ng µg$^{-1}$, indicating that heterogeneous uptake of $H_2O_2$ could account for 86 % of the measured level of $H_2O_2$ in the aerosol phase.

### 3.2.4 Decomposition of organic peroxides

It was found that the concentration of $H_2O_2$ in the extracted solution first increased rapidly, then reached peaks at distinct hours that depended on the particular sample, and finally gradually declined over time. However, interestingly, there was large

sample-to-sample variation, with samples classifiable into three types in terms of the change trend and evolution duration (Fig. 6 and Table 4). The third type (Fig. 6c) occurs when $H_2O_2$ level exhibits a steady decline from 0.03 µM without a growth stage within 13 h, and this was the case with samples 5 and 6 on a slightly polluted day on 2 January 2019. Samples 1 and 2 on 29 December 2018 during clear days belonged to the first type (Fig. 6a), in which $H_2O_2$ rapidly grew within 5 h and subsequently decreased at a slow rate over 25 h. The evolution trends of $H_2O_2$ in the second type (samples 3 and 4, Fig. 6b) during clear

days from 31 December 2018 to 1 January 2019 were similar to the those of the first type, except $H_2O_2$ approached its peak at about 40 h over the whole analysis process lasting for about 300 h.

To seek the reasons for the elevated level of $H_2O_2$ in the extracted solution, we compared the ratio of the maximum ($C_{max}$) to initial ($C_0$) $H_2O_2$ concentration in the extracted solution with the molar concentration ratio of the aerosol-phase TPOs to $H_2O_2$ and found that the ratios of $C_{max}/C_0$ and TPOs/$H_2O_2$ were the same order of magnitude for the first and second types, as

exhibited in Table S5 in the Supplement. This result provided evidence that part of aerosol-phase $H_2O_2$ originated from the decomposition/hydrolysis of organic peroxides, as described in earlier studies (Wang et al., 2011; Li et al., 2016). In the second type, the concentration of TPOs normalized to aerosol mass reached a maximum, indicating that the second type has more TPOs sources and consequently causing higher TPOs/$H_2O_2$ and $C_{max}/C_0$ ratios compared with the first type. Furthermore, the aerosol surface is semi-liquid or liquid under high RH (Liu et al., 2017), which provides reaction sites for the

decomposition/hydrolysis of aerosol-phase organic peroxides. Aerosol-phase organic peroxides could decompose into $H_2O_2$ when the particle aerosols were collected (Zhao et al., 2018). The rates of the decomposition/hydrolysis of organic peroxides to $H_2O_2$ in the first and second types were 0.14 ng µg$^{-1}$ and 3.65 ng µg$^{-1}$, respectively.

The three types of samples were in accordance with the growth process of PM$_{2.5}$. According to meteorological parameters and trace gases data (Table S6 in the Supplement), static weather conditions gradually formed and were accompanied by lower

wind speed, lower ozone level, higher RH, and higher gaseous pollutants. The mass concentration of PM$_{2.5}$ increased from 13.45 µg m$^{-3}$ to 63.11 µg m$^{-3}$. In addition, the mass concentration of TPOs showed a rising trend. But the level of TPOs normalized to aerosol mass increased at first and decreased afterwards due to the rapid growth of PM$_{2.5}$. Because of the consumption of reactive TPOs which formed $SO_4^{2-}$ during polluted days, the rest of the TPOs were stable organic peroxides that could not easily decompose into $H_2O_2$, e.g., peroxide esters (ROOR). The ratio of TPOs/$H_2O_2$ in the third type, collected





on a slightly polluted day, was close to that in the second type on clear days, but a rising trend of $H_2O_2$ in the extracted solution could not be observed. It was calculated that the ratios of decomposable TPOs to total TPOs for the three types were 29 %, 98 %, and 0 %, respectively.

Recently, it was reported that organic peroxides account for a large proportion of secondary organic aerosol (SOA) mass, varying widely from less than 20 % to 60 % (Docherty et al., 2005; Li et al., 2016; Gong et al., 2018). Peroxy radicals also

paly important parts in the formation of highly oxygenated molecules (HOMs) via an autoxidation mechanism, which can form aerosols without sulfuric acid nucleation (Kirkby et al., 2016). The thermal decomposition of peroxide-containing SOAs and HOMs contribute to the formation of aerosol-phase $H_2O_2$ (Krapf et al., 2016). A similar phenomenon was also found by Li et al. (2016), in which the decomposition/hydrolysis of organic peroxides sustainably generated $H_2O_2$ accompanied by the attenuation of TPOs in the extracted solution, and about 18 % of gaseous organic peoxides experienced the heterogeneous

decomposition on aerosols into $H_2O_2$. The decomposable organic peroxides are often peroxycarboxylic acids (PCAs, e.g., peroxyacetic acid, PAA; peroxyformic acid, PFA) and α-hydroxyalkyl-hydroperoxides (α-HAHPs, e.g., hydroxymethyl hydroperoxide, HMHP). Based on previous studies and the evolution trend of the three types in this study, we speculate that PCAs and α-HAHPs accounted for a large proportion of the first and second types, while ROOR played a large part in the third type, as shown in Table 4.

**3.3 Source and sink of $H_2O_2$ in rainwater and aerosol**

To provide support for the sources suggested above, we analysed the source and sink of liquid- and aerosol-phase $H_2O_2$ in rainwater and aerosols. In this study, the measured level of $H_2O_2$ was the concentration after partial or complete reaction with reduced substances, such as $SO_2$ onto the particles. The contribution of different additional sources in the liquid and aerosol phases should be estimated compared with the important sink.

The liquid-phase $H_2O_2$ level was the result of the combined effect between sources (gas-phase partitioning and residual $H_2O_2$ in raindrops) and sinks (reaction with S(IV) and the decomposition of $H_2O_2$). Based on the foregoing description, the dissolved gas-phase $H_2O_2$ in rainwater was 25.20 μM at 298 K. The residual $H_2O_2$ in raindrops could enhance the level of liquid-phase $H_2O_2$ by up to 48.81 μM, based on Henry's law. The largest removal pathway was the consumption by its oxidizing dissolved $SO_2$ into sulfate. Given that the major oxidants to sulfate formation were only $H_2O_2$ and $O_3$ (Penkett et al., 1979; Chandler et

al., 1988), the proportions of the $H_2O_2$ oxidation pathway to the overall, calculated by Eqs. (S12) and (S13) in the Supplement, were 92 % at pH 5 and 11 % at pH 6, respectively. The average of sulfate concentration in rainwater measured 31.95 μM, and the $H_2O_2$ oxidation pathway contributed to the sulfate with 29 μM at pH 5 and 4 μM at pH 6, which was the consumption level of $H_2O_2$. In addition, the decomposition of $H_2O_2$ during 6 h storage time before analysis was 6 μM (Li et al., 2016). To summarize, the concentration of liquid-phase $H_2O_2$ was supposed to have its maximum at 64.01 μM, a bit lower than the 90[th]

percentile of the measured level (67.85 μM). This could be considered to achieve the approximate balance between sources and sinks in the liquid-phase $H_2O_2$. Consequently, the residual $H_2O_2$ in raindrops could explain the difference between $H_A^m$





and $H_A^t$.

In terms of the sources and sinks for aerosol-phase $H_2O_2$, the main removal pathway was the consumption of $H_2O_2$ to sulfate formation, similar to the sink of $H_2O_2$ in the liquid phase. The average mass concentrations of $PM_{2.5}$ and aerosol-phase $SO_4^{2-}$

were 39.21 µg m$^{-3}$ and 2.20 µg m$^{-3}$, respectively. The mass concentration ratio of $SO_4^{2-}$ to $PM_{2.5}$ was 6 % in this study, which was lower than previous studies (Ho et al., 2016; Shao et al., 2018). The discrepancy may be accounted by the decreased ratio of $SO_4^{2-}$ to $PM_{2.5}$ due to $SO_2$ emissions control in recent years, as shown in the Supplement. Accordingly, the $H_2O_2$ oxidation pathway accounted for 57 % of the sulfate formation in a typical haze event on 2−3 January 2019, and the consumption of aerosol-phase $H_2O_2$ caused by sulfate formation maximized at 11.33 ng µg$^{-1}$. In this study, the maximum amount of $H_2O_2$

formed by the decomposition/hydrolysis of organic peroxides was close to 3.65 ng µg$^{-1}$, accounting for 32 % of the $H_2O_2$ formation in the aerosol phase. The heterogeneous uptake of $H_2O_2$ on aerosols had a minor contribution to the aerosol-phase $H_2O_2$, and the proportion was less than 0.5 %. The sources could not achieve a balance with the consumption of the aerosol-phase $H_2O_2$. Provided that the $\gamma$ value could reach $10^{-3}$ (Wang et al., 2016), the amount of heterogeneous uptake could reach 0.49 ng µg$^{-1}$. However, this still does not bridge the difference between sinks and sources. In our view, there are a couple of

possible explanations for the difference. First, we estimated the contribution of the $H_2O_2$ oxidation pathway to sulfate formation during the entire measurement period based on the contribution of high-pollution days, which may overestimate the sink for the aerosol-phase $H_2O_2$. Provided that the contribution ratio was 20 % on clear days, the sink for aerosol-phase $H_2O_2$ to sulfate formation could be 3.97 ng µg$^{-1}$, which was in general accordance with the sources for aerosol-phase $H_2O_2$. Second, the inverse dependence of the $\gamma$ value on the gas-phase $H_2O_2$ concentration was not considered, and the $\gamma$ value could have been

underestimated when the level of gas-phase $H_2O_2$ in winter was much lower (Romanias et al., 2012; Romanias et al., 2013). Third, there may be missing sources in aerosol-phase $H_2O_2$. There are possibly some potential sources that are not completely understood, such as the heterogeneous uptake of $HO_2$ on aerosols forming aerosol-phase $H_2O_2$ (Liang et al., 2013) and so on. Based on the above analysis, sources and sinks of $H_2O_2$ in the liquid phase could achieve balance, while the formation of $H_2O_2$ from the decomposition/hydrolysis of aerosol-phase organic peroxides and the heterogeneous uptake of $H_2O_2$ could not offset

the consumption of $H_2O_2$ in the aerosol phase. Two-thirds of the total aerosol-phase $H_2O_2$ formation failed to be explained by the two possible sources. Field measurements and laboratory experiments are urgently needed to further study the possible reasons and search for other potential sources of aerosol-phase $H_2O_2$.

## 4 Conclusions

In this study, we simultaneously measured $H_2O_2$ concentrations in gas and rainwater in summer as well as in the gas and aerosol

phases ($PM_{2.5}$) in winter over urban Beijing. For the investigated seven rain episodes, the average $H_A^m$ was $2.1 \times 10^5$ M atm$^{-1}$, which was 2.5 times greater than $H_A^t$ at 298 ± 2 K. The liquid-phase concentration of $H_2O_2$ averaged 44.12 ± 26.49 µM. In 77 % of the rain samples, the liquid-phase $H_2O_2$ level was much larger than the predicted values estimated for pure water using Henry's law. We found that 12 % of measured $H_2O_2$ in all samples and 54 % of measured $H_2O_2$ in those samples that did not



follow Henry's law were from residual $H_2O_2$ in raindrops. With an increase in raindrops diameter and fall distance, the proportion of the additional source of liquid-phase $H_2O_2$ gradually increased. Furthermore, the source and sink in rainwater could achieve a balance.

For the measured $PM_{2.5}$ aerosol samples, a similar phenomenon was observed between the measured and predicted levels of $H_2O_2$ in the aerosol phase, but the difference was much higher than that in the liquid phase. $K_P^m$ averaged $3.8 \times 10^{-3}$ $m^3$ $\mu g^{-1}$, which was four orders of magnitude higher than $K_P^t$ at $270 \pm 4$ K. The aerosol-phase concentration of $H_2O_2$ normalized to the aerosol mass averaged $0.093 \pm 0.085$ ng $\mu g^{-1}$. Aerosol-phase $H_2O_2$ level was associated with relative humidity, increasing before decreasing, which resulted from the competition between the formation and consumption of $H_2O_2$. The decomposition/hydrolysis of organic peroxides produced the elevated aerosol-phase $H_2O_2$ at a maximum rate of 3.65 ng $\mu g^{-1}$, which was responsible for 32 % of the formation of aerosol-phase $H_2O_2$. The heterogeneous uptake of $H_2O_2$ played a minor role in increasing the $H_2O_2$ level in the aerosol phase, and the proportion was less than 0.5 %. There are many uncertainties in the decomposition/hydrolysis of organic peroxides in this study, and laboratory simulation studies are needed to quantify the roles of different organic peroxides in the decomposition process. Aerosol-phase $H_2O_2$ in this study cannot reach source and sink equilibrium, and there are missing sources of aerosol-phase $H_2O_2$. Due to a lack of substantial severe haze events with high RH in this study, the source and sink mentioned in the aerosol-phase $H_2O_2$ need to be further verified.

Our study has provided direct evidence to prove that the partitioning of $H_2O_2$ between the gas-liquid and gas-aerosol phases not only follows thermodynamic equilibrium but is affected by certain physical and chemical reactions. The effective field-derived Henry's law constant and gas-aerosol partitioning coefficient should be accepted to better predict the measured liquid- and aerosol-phase $H_2O_2$ concentrations, which would be beneficial to correctly calculating the contribution of $H_2O_2$ to the fast formation of $SO_4^{2-}$ and $PM_{2.5}$ during pollution episodes. More laboratory experiments and field measurements are urgently needed to improve our understanding of the partitioning of peroxides in different phases in the atmosphere.

*Data availability.* The data are accessible by contacting the corresponding author (zmchen@pku.edu.cn).

*Author contributions.* In the framework of BJ-2018Summer and BJ-2018Winter measurements, ZC and XX designed the study. XX carried out all peroxide measurements used in this study, analysed the data, and wrote the paper. ZC helped interpret the results, guided the writing, and modified the manuscript. YG contributed to the methods of analysing aerosol-phase hydrogen peroxides and total peroxides. HS helped interpret data and modify the paper. SC provided the data for the meteorological parameters, trace gases and $PM_{2.5}$ mass concentrations. All authors discussed the results and contributed to the final paper.

*Competing interests.* The authors declare that they have no conflict of interest.

*Acknowledgements.* This work was funded by the National Key Research and Development Program of China (Grants 2016YFC0202704), National Research Program for Key Issues in Air Pollution Control (Grants DQGG0103) and the National



Natural Science Foundation of China (Grants 21477002).





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







**Figure 1: Time profiles of measured and predicted concentrations of H₂O₂ from seven rain episodes. The seven rainfall events are listed in chronological order: (a) 24 July, (b) 25 July, (c) 5 August, (d) 6 August, (e) 8 August, (f) 30 August, and (g) 2 September.**





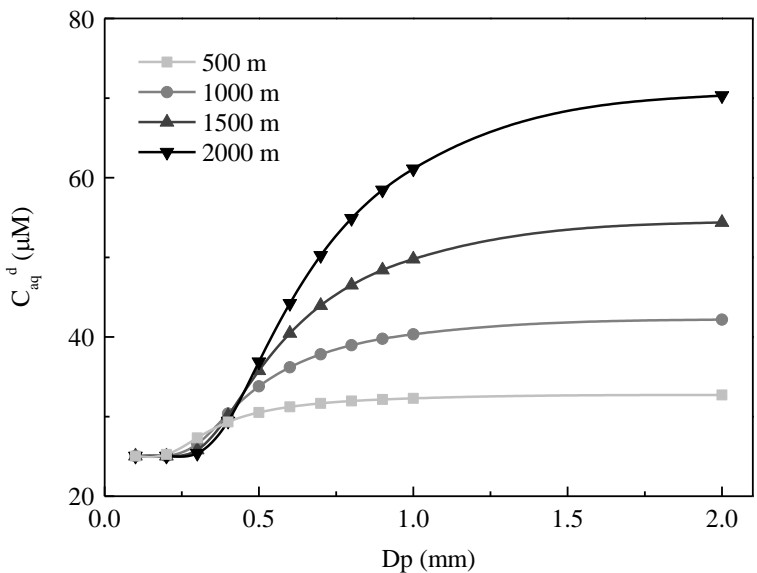

**Figure 2: The dependence of the concentration of H₂O₂ in the ground raindrops ($C_{aq}^{d}$) on the diameter of the raindrops. The light grey, grey, dark grey, and black lines denote fall distances of 500 m, 1000 m, 1500 m, and 2000 m, respectively.**






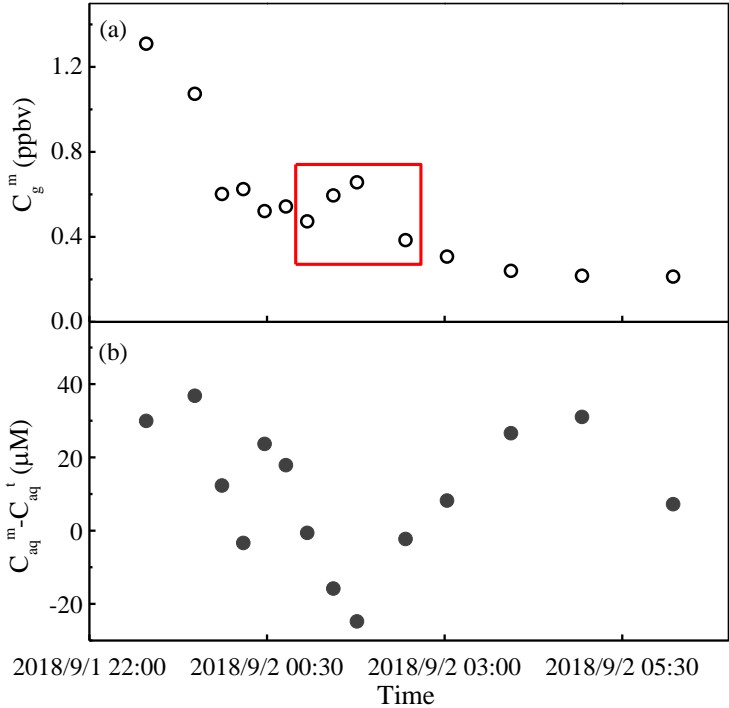

**Figure 3: The measured and predicted H₂O₂ levels in a rain event on 1−2 September 2018. (a) Gas-phase H₂O₂ ($C_g^m$). (b) The difference between measured ($C_{aq}^m$) and theoretical ($C_{aq}^t$) levels of H₂O₂ in the liquid phase. The red box indicates a sudden rise in gas-phase H₂O₂.**






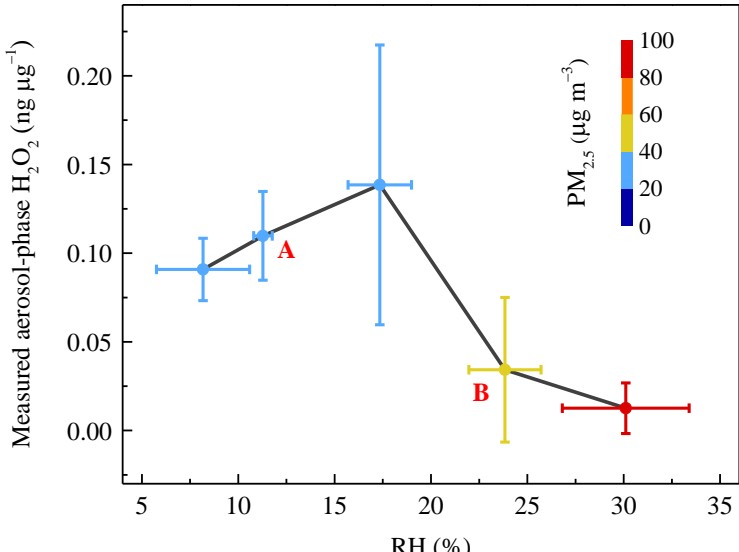

**Figure 4: The relationship between measured aerosol-phase H₂O₂ level and relative humidity. Coloured circles denote the mass concentration of PM₂.₅. The vertical error bars represent the standard deviations of aerosol-phase H₂O₂ concentration in every RH range bin.**






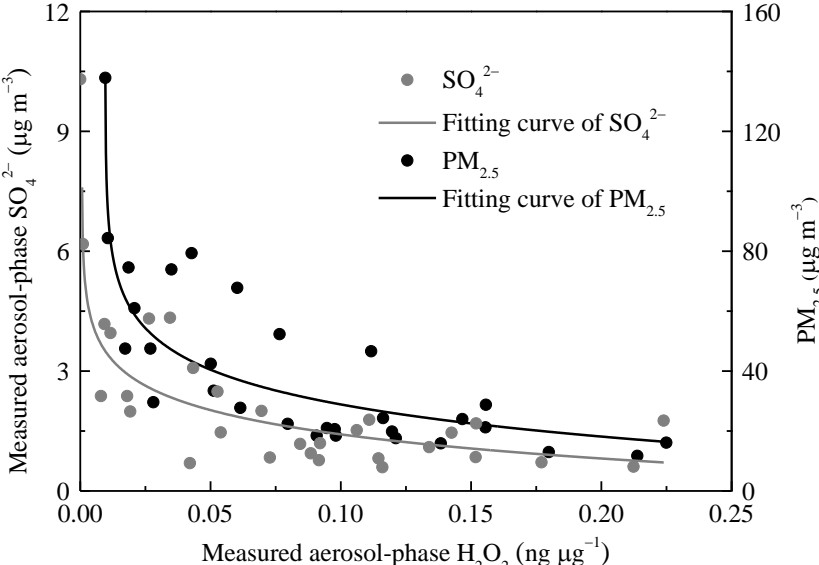

**Figure 5: The negative dependence of the measured concentrations of aerosol-phase SO$_4^{2-}$ and PM$_{2.5}$ on H$_2$O$_2$. The grey and black lines are the logarithmic fits for SO$_4^{2-}$ level in aerosols and the PM$_{2.5}$ mass concentration, respectively.**




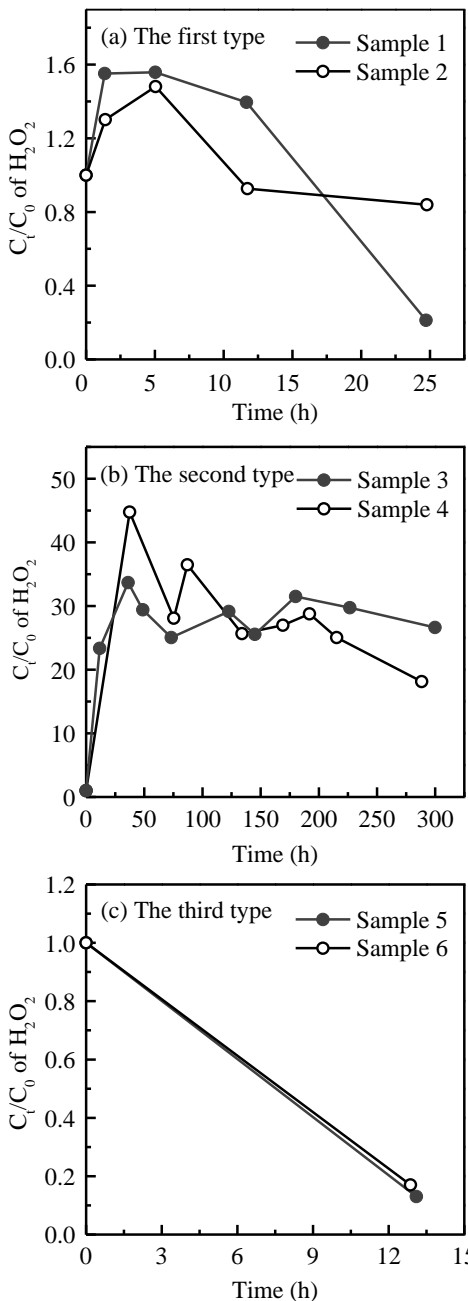

**Figure 6: Time profiles of aerosol-phase H₂O₂ evolution in the extracted solution. (a) The first type: samples 1 and 2 were collected on 29 December 2018. (b) The second type: samples 3 and 4 were gathered on 31 December 2018−1 January 2019. (c) The third type: samples 5 and 6 were collected on 2 January 2019.** $C_t$ **and** $C_0$ **denote molar concentrations of H₂O₂ in the extracted solution at time=t and time =0.**






**Table 1: Calculating the level of H₂O₂ in the ground raindrops ($C_{aq}^d$) with different diameters from a height of 1000 mᵃ.**

| Parameters | $D_P$ (mm) | | | |
|---|---|---|---|---|
| | 0.1 | 0.5 | 1.0 | 2.0 |
| $u$ (m s⁻¹)[b] | 0.27 | 2.06 | 4.03 | 6.49 |
| $k_g$ (cm s⁻¹)[c] | 51.42 | 25.00 | 21.09 | 17.45 |
| $C_{aq}^d$ (μM) | 25.03 | 33.81 | 40.34 | 42.18 |

ᵃ These parameters are calculated based on equations in Gunn and Kinzer (1949), Levine and Schwartz (1982), Kumar (1985), and Seinfeld and Pandis (2006).

ᵇ $u$ is the terminal fall velocity of a raindrop.

ᶜ $k_g$ is the mass transfer coefficient of H₂O₂ in the gas phase.



**Table 2: Estimates of the level of H₂O₂ in cloud water ($C_{aq}^c$) and the surrounding atmosphere ($C_g^c$) at different fall distances with a raindrop diameter of 1.0 mm.**

| Parameters | 500 m | 1000 m | 1500 m | 2000 m |
|---|---|---|---|---|
| $T_S^c$ (K)[a] | 295 | 292 | 289 | 286 |
| $H_A^t$ (M atm$^{-1}$) | $1.1 \times 10^5$ | $1.4 \times 10^5$ | $1.9 \times 10^5$ | $2.5 \times 10^5$ |
| $C_{aq}^c$ (μM) | 45.64 | 47.27 | 49.04 | 50.95 |
| $C_g^c$ (ppbv) | 0.41 | 0.33 | 0.26 | 0.21 |

[a] $T_S^c$ is the temperature in cloud water.





**Table 3: Comparison between the theoretical heterogeneous uptake of H₂O₂ on aerosols ($[X]_p^{t,h}$) and the measured aerosol-phase H₂O₂ level ($[X]_p^m$)ᵃ.**

| Parameters | $T_W$ (K) | $RH$ (%) | $\gamma$ —ᵇ | $A_{es}$ (cm²)ᶜ | $[X]_g$ (molecules m⁻³)ᵈ | $[X]_p^{t,h}$ (ng µg⁻¹) | $[X]_p^m$ (ng µg⁻¹) |
|---|---|---|---|---|---|---|---|
| Averages | 270 | 17.89 | $1.54 \times 10^{-4}$ | 9.00 | $6.65 \times 10^{14}$ | 0.049 | 0.057 |

ᵃ These parameters are calculated based on Wu et al. (2015).
ᵇ $\gamma$ is the heterogeneous uptake coefficient, dimensionless.
ᶜ $A_{es}$ is the effective reaction area of aerosols.
ᵈ $[X]_g$ is the concentration of gas-phase H₂O₂.






**Table 4: Comparison of the H$_2$O$_2$ evolution parameters in the extracted solution among the three types.**

| Parameters | First type | Second type | Third type |
|---|---|---|---|
| Peak time (h) | 5 | 40 | − |
| Decomposition rate of organic peroxides to H$_2$O$_2$ (ng µg$^{-1}$) | 0.14 | 3.65 | − |
| $C_{max}/C_0$ of H$_2$O$_2$ (µM/µM) | 1.52 | 39.22 | 1.00 |
| TPOs/H$_2$O$_2$ (µM/µM) | 5.25 | 40.06 | 47.59 |
| Ratio of decomposable organic peroxides (%) | 29 | 98 | 0 |
| Possible organic peroxides | PCAs[a], α-HAHPs[b] | PCAs, α-HAHPs | ROOR[c] |

[a] PCAs denote peroxycarboxylic acids, e.g., peroxyacetic acid, PAA; peroxyformic acid, PFA.

       [b] α-HAHPs denote α-hydroxyalkyl-hydroperoxides, e.g., hydroxymethyl hydroperoxide, HMHP.

       [c] ROOR denote peroxide esters.