# Peer review of "Partitioning of hydrogen peroxide in gas-liquid and gas-aerosol phases"

_Atmospheric Chemistry and Physics, 2020_

## Referee Comment (RC1) · Anonymous Referee #1 · 17 Feb 2020

General comments: Xuan et al. performed field measurements of the gas-, liquid- and aerosol-phase $H_2O_2$ in the urban atmosphere of Beijing to understand the partitioning of $H_2O_2$ between gas- and liquid-phase or aerosol-phase. They show that the partitioning of $H_2O_2$ in the gas-liquid phase can be explained by Henry's law and the residual $H_2O_2$ in the raindrops while the aerosol-phase $H_2O_2$ level is significantly higher than that predicted value based on Pankow's absorptive partitioning theory. This paper has important implications for understanding the $H_2O_2$ chemistry and sulfate formation in the atmosphere, so it is well within the scope of ACP. This paper is of great interest to the atmospheric community although some clarifications regarding the data analysis are required. I recommend this paper to be published after addressing the specific comments below.

[Figure]

Specific comments: Estimation of effective partitioning coefficients: The authors determined the gas-aerosol portioning coefficient instead of the effective Henry's law constant for the gas-aerosol phase. Is this due to that aerosol water content can not be accurately estimated for low RH? The effective Henry's law constant should be estimated for the high RH condition, e.g. heavy haze episodes from 2 Jan to 3 Jan 2019 and compared with the theoretical value.

Sources and sink of $H_2O_2$ in aerosol: 1) The authors estimated that heterogeneous uptake of $H_2O_2$ could account for 86% of the measured $H_2O_2$ in the aerosol phase in Sec 3.2.3 while stated that the heterogeneous uptake of $H_2O_2$ on aerosols contributed less than 0.5% of the aerosol-phase $H_2O_2$ in Sec 3.3. Please clarify.

2) The authors stated that the rates of the decomposition/hydrolysis of organic peroxides in the first and second types were 0.14 ng ug-1 and 3.65 ng ug-1 (lines 296-297) and further estimated the contribution of decomposition/hydrolysis of organic peroxides to aerosol $H_2O_2$ to be 32% (lines 343-346). However, these numbers seem to be the steady-state or maximum amount of $H_2O_2$, not formation rates. The estimation should be based on the formation and consumption rate of $H_2O_2$.

3) Though the heterogeneous uptake of $HO_2$ on aerosols is not well understood, it is possible to estimate its contribution to aerosol $H_2O_2$ using the reactive uptake coefficient of $HO_2$ to aerosol from literature and assuming the product to be $H_2O_2$ (Li et al., 2019). It is recommended to perform such calculations to provide more insights.

4) The authors should discuss the "salting in" effect of high ionic strength of aerosol particles on gas-aerosol partitioning of $H_2O_2$ though it may only have a minor contribution to the enhanced aerosol $H_2O_2$ concentrations.

Line 82: Are the organic peroxide concentrations corrected for the collection efficiency?

Lines 149-150: Please explain how 88% is derived.

Line 181: What is the gas-phase $H_2O_2$ concentration used to estimate the liquid-phase

H2O2?

Section 3.2.4: The experimental details on the decomposition of organic peroxides should be provided. Is the extracted solution exposed to light at room temperature? Are these experiments conducted at atmospheric relevant conditions so that the derived rates of decomposition can be applied to ambient?

Technical corrections:

Lines 59-60: References are missing.

Equation 4: TSP or PM2.5 should be used instead of Com.

Line 331: "measured" should be "was measured to be".

References: Li, K., Jacob, D.J., Liao, H. et al. A two-pollutant strategy for improving ozone and particulate air quality in China. Nat. Geosci. 12, 906–910 (2019). https://doi.org/10.1038/s41561-019-0464-x.

————————————————

---

## Referee Comment (RC2) · Anonymous Referee #2 · 18 Feb 2020

This manuscript by Xuan et al describes a detailed study of the partitioning of $H_2O_2$ in the atmosphere through field measurements. The authors quantified $H_2O_2$ in the gas phase, aerosol, and rainwater (as a surrogate for cloudwater). By comparing the measured and theoretical Henry's law constant, as well as the measured and theoretical partitioning coefficient, the authors conclude that the measured values for both are higher than the theoretical values. An in-depth assessment is conducted to evaluate the influence of raindrop falling on the quantified $H_2O_2$ concentration, as well, a discussion on the source and sink of $H_2O_2$ in aerosol is provided. $H_2O_2$ plays an important role in the atmosphere, and understanding its partitioning in different atmospheric phases is of great importance for the atmospheric chemistry community. The manuscript is within the scope of ACP. The data analysis and calculation were per-

formed with caution. I recommend publication on ACP after addressing the following comments.

Major comment: In section 3.2.4, the authors present the evolution of H2O2 as a function of time in the aerosol abstract, and a detailed discussion on the potential source of H2O2. This result highlights the challenges in making off-line H2O2 measurement from filter samples. Especially, when the sampling time is as long as 11.5 h (Line 100), it is very likely that the organic peroxides present in the aerosol sample is continuously decomposing on the filter. The authors categorize the H2O2 evolution into three types and postulate the relevant source of H2O2 for each type. However, in my opinion, this appears too speculative. The decomposition of H2O2 on filter is difficult to control, and the quantified H2O2 could be merely a snapshot of an ongoing decomposition process. The authors must justify whether it is valid at all to establish gas-aerosol partitioning of H2O2 based on the current technique.

Minor comments:

- Literature-reported Henry's law constants of H2O2 varies across a certain range. The authors should justify why they used 8.4x104 M/atm. Is this the recommended value by the JPL publication?

- Line 139 – Is the PM2.5 concentration a good indicator for Com in a polluted environment like Beijing?

- Line 150 – The authors state that when HmA was less than HtA, the samples followed Henry's law. Why? Shouldn't they agree (neither higher nor lower)?

- Line 275 – The authors report here that heterogeneous uptake can count for 86% of aerosol phase H2O2. Later in Line 346, the author report 0.5%. Please clarify.

- Line 370 – correct me if I am wrong. "additional source of liquid-phase H2O2 gradually increased" – should this be the sink of H2O2 due to droplet-to-gas transfer is gradually reduced?

- Figure 5, and Line 263 – the authors interpret the inversely related H2O2 concentration and PM2.5/sulfate concentrations as a result of a H2O2 sink by SO2 oxidation. However, could the inversed relation be just due to dilution of H2O2 when aerosol loading is high?

Technical comments:

- Line 59- "easily to absorbed" to "easily absorbed"

- Line 144 – "statistically counted" appears awkward. Should probably remove.

- Line 146 – "with" 25.20 $\mu$M – is 25.20 $\mu$M the theoretical value? "with" makes the sentence unclear.

- Line 171 – "almost" less than 2000 m – should this be "always" less than 2000m?

---

## Author Comment (AC1) · 20 Mar 2020

**Response to Reviewer #1**

We gratefully thank you for your constructive comments and thorough review. Our point-by-point responses can be found below.

(Q=Question, A=Answer, C=Change in the revised manuscript)

**General comments:**

Q1: Xuan et al. performed field measurements of the gas-, liquid- and aerosol-phase $H_2O_2$ in the urban atmosphere of Beijing to understand the partitioning of $H_2O_2$ between gas- and liquid-phase or aerosol-phase. They show that the partitioning of $H_2O_2$ in the gas-liquid phase can be explained by Henry's law and the residual $H_2O_2$ in the raindrops while the aerosol-phase $H_2O_2$ level is significantly higher than that predicted value based on Pankow's absorptive partitioning theory. This paper has important implications for understanding the $H_2O_2$ chemistry and sulfate formation in the atmosphere, so it is well within the scope of ACP. This paper is of great interest to the atmospheric community although some clarifications regarding the data analysis are required. I recommend this paper to be published after addressing the specific comments below.

A1: We highly appreciate your comments and suggestions. The questions you mentioned are specifically answered as follows.

**Specific comments:**

Q2. Estimation of effective partitioning coefficients: The authors determined the gas-aerosol portioning coefficient instead of the effective Henry's law constant for the gas-aerosol phase. Is this due to that aerosol water content cannot be accurately estimated for low RH? The effective Henry's law constant should be estimated for the high RH condition, e.g. heavy haze episodes from 2 Jan to 3 Jan 2019 and compared with the theoretical value.

A2: Yes, you are right. After considering your suggestion, we have calculated the effective Henry's law constant for the gas-aerosol phase during a heavy haze episode from 2 Jan to 3 Jan 2019 and compared it with the theoretical value in the revised manuscript.

C2: Lines 267−272 in Sec. 3.2.1:

"Because aerosol water content (AWC) cannot be correctly evaluated at low RH, the effective field-derived Henry's law constant ($H_P^m$) of $H_2O_2$ was estimated for high RH condition, e.g. a heavy haze

episode from 2 January to 3 January 2019 (RH, 30 %). Details regarding the estimation of AWC was shown in the Supplement. It was calculated that AWC, $C_p^m$ and $C_g^m$ levels averaged 3.20 μg m$^{-3}$, 6.63 × 10$^3$ μM, and 1.90 × 10$^{-11}$ atm. Based on Eq. (5), the average $H_P^m$ on 2−3 January 2019 was calculated to be 2.7 × 10$^8$ ± 1.8 × 10$^8$ M atm$^{-1}$. However, the theoretical Henry's law constant ($H_P^t$) at 270 K was 1.1 × 10$^6$ M atm$^{-1}$ (Sander et al., 2011), which was lower than $H_P^m$ by two orders of magnitude."

Q3. The authors estimated that heterogeneous uptake of H$_2$O$_2$ could account for 86% of the measured H$_2$O$_2$ in the aerosol phase in Sec 3.2.3 while stated that the heterogeneous uptake of H$_2$O$_2$ on aerosols contributed less than 0.5% of the aerosol-phase H$_2$O$_2$ in Sec 3.3. Please clarify.

A3: Thanks for your advice. The two percentages are calculated in different methods. 86 % refers to the ratio of the amount of heterogeneous uptake of H$_2$O$_2$ to the measured aerosol-phase H$_2$O$_2$ level, while 0.5 % refers to the ratio of the amount of heterogeneous uptake of H$_2$O$_2$ to the consumption amount of aerosol-phase H$_2$O$_2$. In addition, we have revaluated the contribution of the heterogeneous uptake to the aerosol-phase H$_2$O$_2$ based on the formation and consumption rates according to the reviewers' suggestions, and the heterogeneous uptake could account for 2 % of the consumption rate of the aerosol-phase H$_2$O$_2$. To avoid confusion, we have removed 86 % in Sec. 3.2.3 and 0.5 % in Sec. 3.3.

Q4. The authors stated that the rates of the decomposition/hydrolysis of organic peroxides in the first and second types were 0.14 ng μg$^{-1}$ and 3.65 ng μg$^{-1}$ (lines 296−297) and further estimated the contribution of decomposition/hydrolysis of organic peroxides to aerosol H$_2$O$_2$ to be 32% (lines 343−346). However, these numbers seem to be the steady-state or maximum amount of H$_2$O$_2$, not formation rates. The estimation should be based on the formation and consumption rate of H$_2$O$_2$.

A4: Thanks for your suggestion. We have recalculated the estimation considering the formation and consumption rates of H$_2$O$_2$ and removed the calculation based on the steady-state or maximum amount of H$_2$O$_2$ in the revised manuscript. Furthermore, we have changed the relevant data in Table 3 and 4.

C4: Lines 417−430 in Sec. 3.3:

"We estimated the contribution of different sources to the aerosol-phase H$_2$O$_2$ based on the formation and consumption rates. According to the previous estimation of the theoretical sulfate formation rate from January 2 to January 3 2019 (0.29 μg m$^{-3}$ h$^{-1}$) and the average mass concentration of PM$_{2.5}$ (106.19

μg m$^{-3}$), the consumption rate of H$_2$O$_2$ should be 0.97 ng μg$^{-1}$ h$^{-1}$. With respect to the sources of the aerosol-phase H$_2$O$_2$, the decomposition/hydrolysis of organic peroxides was firstly considered, with average rates of the rising stage for the first and second types (Fig. 6), 0.01 ng μg$^{-1}$ h$^{-1}$ and 0.10 ng μg$^{-1}$ h$^{-1}$, respectively. Because the extracted solution was stored under 255 K, lower than the actual atmospheric temperature (270 K), the decomposition/hydrolysis rates of organic peroxides were underestimated and an adjusting factor should be multiplied. The factors for the three typical labile organic peroxides (HMHP, PFA, and PAA) were 13, 3, and 2, respectively, as shown in the Supplement. Assuming the factor was in the range of 2−13, the average decomposition/hydrolysis rate of organic peroxides for the first and second types (0.055 ng μg$^{-1}$ h$^{-1}$) was used to calculate the formation rate. The formation rate of the aerosol-phase H$_2$O$_2$ from the decomposition/hydrolysis of organic peroxides could account for 11−74 % of the consumption rate by sulfate formation. Moreover, the heterogenous uptake of HO$_2$ and H$_2$O$_2$ were also likely to improve the aerosol-phase H$_2$O$_2$ level at the rates of 0.22 ng μg$^{-1}$ h$^{-1}$ and 0.02 ng μg$^{-1}$ h$^{-1}$, respectively, which can offset 22 % and 2 % of the consumption rate of H$_2$O$_2$, respectively."

Q5. Though the heterogeneous uptake of HO$_2$ on aerosols is not well understood, it is possible to estimate its contribution to aerosol H$_2$O$_2$ using the reactive uptake coefficient of HO$_2$ to aerosol from literature and assuming the product to be H$_2$O$_2$ (Li et al., 2019). It is recommended to perform such calculations to provide more insights.

A5: Thanks for your advice. We have added the calculation of the heterogeneous uptake of HO$_2$ on aerosols in the revised manuscript.

C5: Lines 325−331 in Sec. 3.2.3:

"As HO$_2$ radical is a precursor of H$_2$O$_2$, the heterogeneous uptake of HO$_2$ onto aerosols may also contribute to the formation of the aerosol-phase H$_2$O$_2$. We assumed that the reactive uptake coefficient of HO$_2$ to aerosol particles was 0.2, and the product of HO$_2$ was H$_2$O$_2$ (Li et al., 2019). At the same observation site in winter of 2017, HO$_2$ concentration for noontime averaged $(0.4 \pm 0.2) \times 10^8$ cm$^{-3}$ and $(0.3 \pm 0.2) \times 10^8$ cm$^{-3}$ on clean and polluted days, respectively (Ma et al., 2019). Since HO$_2$ level data in 2018 was not available, we used the level of HO$_2$ on clean days in winter of 2017 for calculations, and the average was about $0.2 \times 10^8$ cm$^{-3}$ at day-time. The heterogenous uptake rate of HO$_2$ on aerosols was calculated the same way as H$_2$O$_2$, and the formation rate of the aerosol-phase H$_2$O$_2$ by reactive uptake of

HO$_2$ averaged 0.22 ng $\mu$g$^{-1}$ h$^{-1}$ at all day.”

Q6. The authors should discuss the "salting in" effect of high ionic strength of aerosol particles on gas-aerosol partitioning of H$_2$O$_2$ though it may only have a minor contribution to the enhanced aerosol H$_2$O$_2$ concentrations.

A6: Thanks for your suggestion. We have discussed the "salting-in" effect of high ionic strength of aerosol particles on the gas-aerosol partitioning of H$_2$O$_2$ in the revised manuscript.

C6: Lines 272−278 in Sec. 3.2.1:

"In Chung's study (2005), "salting-in" effect can improve the level of H$_2$O$_2$ by a factor of two when the concentrations in salt solutions were up to 10 M, and the most obvious "salting-in" effect of salt solutions was ammonium sulfate. In this paper, the levels of aerosol-phase NH$_4^+$ and SO$_4^{2-}$ on 2−3 January 2019 were 94 M and 21 M, respectively, and the level of (NH$_4$)$_2$SO$_4$ was assumed to be 21 M. The increasement of $H_P^m$ by the "salting-in" effect of (NH$_4$)$_2$SO$_4$ was about 3.2 × 10$^6$ M atm$^{-1}$ at 286 K based on equations in Chung et al. (2005). Even though aerosol particles were collected at 270 ± 4 K and the increasement may be greater, the "salting-in" effect could not fully explain the difference between $H_P^m$ and $H_P^t$. Other sources need to be found later."

Q7. Line 82: Are the organic peroxide concentrations corrected for the collection efficiency?

A7: Thanks for your suggestion. Because the measured concentrations of organic peroxides were near the detection limit of the HPLC method in BJ-2018Winter measurement, the levels of organic peroxides were not discussed in this paper. Alternatively, we used the concept of total peroxides as a measure to estimate the sources of the aerosol-phase H$_2$O$_2$ in Sec. 3.2. The level of total peroxides was measured using the iodometric spectrophotometric method with an extraction efficiency close to ~ 98 % (Li et al., 2016), so we did not correct the total peroxides level.

Q8. Lines 149−150: Please explain how 88% is derived.

A8: Thanks for your suggestion and we have explained it in line 166 in the revised manuscript. The percentage was calculated as the average ratio of the predicted level to measured level of H$_2$O$_2$ in each rain sample. The value indicated that 88 % of the liquid-phase H$_2$O$_2$ in all the rain samples collected was from gas-phase partitioning. Since the measured H$_2$O$_2$ level in some rain samples was lower than the

predicted value, gas-phase partitioning accounted for a high proportion in all samples based on statistics on averages.

Q9. Line 181: What is the gas-phase $H_2O_2$ concentration used to estimate the liquid-phase $H_2O_2$?.

A9: The gas-phase $H_2O_2$ concentration used to estimate the liquid-phase $H_2O_2$ was $0.30 \pm 0.26$ ppbv, and we have added it in lines 203−204 in the revised manuscript. We assumed that the gas-phase $H_2O_2$ concentration was homogeneous, and the gas-phase $H_2O_2$ in the cloud atmosphere was equal to the gas-phase $H_2O_2$ near the ground (line 191).

Q10. Section 3.2.4: The experimental details on the decomposition of organic peroxides should be provided. Is the extracted solution exposed to light at room temperature? Are these experiments conducted at atmospheric relevant conditions so that the derived rates of decomposition can be applied to ambient?

A10: Thanks for your suggestion. The extracted solution was away from light at 255 K. The experimental conditions were chosen based on certain considerations, which have been added in the revised manuscript and Supplement. The influence of experimental conditions on the derived rates has also been discussed in the revised manuscript and Supplement. We think that the derived rates of decomposition can be applied to the ambient atmosphere.

C10: Lines 111−115 in Sec. 2.2.3:

"The remaining extracted solution was stored at 255 K away from light for subsequent measurement of $H_2O_2$ concentration variation with time, and details of the experimental conditions of the extracted solution are shown in the Supplement. Photochemical reactions of aerosols may produce aerosol-phase $H_2O_2$ (Zhou et al., 2008), and the effect of the photochemical reactions on the level of $H_2O_2$ in the extracted solution was discussed in the Supplement."

Lines 1−48 in the Supplement:

"The extracted solution was stored under refrigeration at 255 K, away from light. The first reason to choose 255 K was that the temperature during BJ-2018Winter measurement averaged 270 K, less than 273 K. The second reason was that under 255 K, the decomposition rate of $H_2O_2$ should be reduced, which contributed to a more accurate estimation of the decomposition rates of organic peroxides to $H_2O_2$. Li et al. (2016) studied the stability of $H_2O_2$ in SOA stored on-filter at 255 K and 298 K. It was found

that the level of $H_2O_2$ remained stable for 6 days at 255 K but decreased gradually at 298 K. The third reason was that the $H_2O_2$ level in the extracted solution was very low at time=0, which could easily decompose at 277 K, therefore, the extracted solution should be stored at 255 K.

Due to the positive correlation between temperature and decomposition rates, the derived rates of decomposition in this paper were lower than the actual rates of decomposition. To discuss the influence of the storage temperature on the decomposition rates of organic peroxides, hydroxymethyl hydroperoxide (HMHP), peroxyformic acid (PFA), and peroxyacetic acid (PAA) were chosen as representatives. According to the Arrhenius equation, the reaction rate usually increases exponentially as temperature increases. The ratios of the decomposition/hydrolysis rates of HMHP, PFA, and PAA at 270 K to 255 K were 13, 3, and 2, respectively (Zhou and Lee, 1992; Dul'neva and Moskvin, 2005; Sun et al., 2011). We have considered the influence of temperature on the decomposition rates of organic peroxides when calculating the aerosol-phase $H_2O_2$ formation rate from the decomposition/hydrolysis of organic peroxides, as shown in Sec. 3.3.

The extracted solution was away from light in this paper, which was different from atmospheric relevant conditions (i.e., exposure to sunlight in the ambient atmosphere), and may affect the data applicability in this study. We chose this experimental condition because if the extracted solution was exposed to sunlight, the photochemical reactions of organic matters and the decomposition/hydrolysis of organic peroxides will coexist, and we cannot distinguish the effects of these two processes. By doing so, the specific contribution of the decomposition/hydrolysis of organic peroxides to the aerosol-phase $H_2O_2$ was estimated. With respect to the photochemical reactions of organic matters, Zhou et al. (2008) have discussed that as the exposure time of the extracted solution to sunlight increased, the production of peroxides in nascent marine aerosols first increased rapidly and then slowly. The change trend in Zhou's study was the same as that of the aerosol-phase $H_2O_2$ level in this paper (Fig. 6). The estimated 24-h-average rate of $H_2O_2$ photochemical production in Alert particles was about 9 mM $h^{-1}$ at 248 K (Anastasio and Jordan, 2004). We assumed that the photoformation rate of $H_2O_2$ in Beijing particles was also 9 mM $h^{-1}$. And the concentrations of AWC and $PM_{2.5}$ from 2 January to 3 January 2019 were 3.20 μg $m^{-3}$ and 90.36 μg $m^{-3}$, respectively. The formation rate of the aerosol-phase $H_2O_2$ from photochemical reactions was estimated to be 0.011 ng μg$^{-1}$ h$^{-1}$ at 248 K. In addition, the activation energy of $H_2O_2$ photoformation was 9 kJ mol$^{-1}$ (Anastasio et al., 1994), and the rate of $H_2O_2$ photoformation at 270 K should be 1.4 times higher than the value at 248 K. Compared with the aerosol-phase $H_2O_2$ formation rates from the

decomposition/hydrolysis of organic peroxides and the heterogeneous uptake of $HO_2$, the $H_2O_2$ photoformation could be neglected.

Based on the above analysis, we believe that the derived rates of decomposition under the experimental conditions in this paper can be applied to the ambient atmosphere."

**Technical corrections:**

Q11. Lines 59−60: References are missing.

A11: Thanks for your reminder. We have added related references in line 65.

Q12. Equation 4: TSP or $PM_{2.5}$ should be used instead of $C_{om}$.

A12: Yes, we have changed "$C_{om}$" into "TSP" in Eq. (4) in the revised manuscript, and used the $PM_{2.5}$ mass concentration as an indicator because the $PM_{2.5}$ concentration was available based on our measurements.

Q13. Line 331: "measured" should be "was measured to be".

A13: Thanks for your advice. We have revised it in line 391.

**References**

Anastasio, C., Faust, B. C., and Allen, J. M.: Aqueous phase photochemical formation of hydrogen peroxide in authentic cloud waters, J. Geophys. Res., 99, 8231−8248, https://doi.org/10.1029/94JD00085, 1994.

Anastasio, C., and Jordan, A. L.: Photoformation of hydroxyl radical and hydrogen peroxide in aerosol particles from Alert, Nunavut: implications for aerosol and snowpack chemistry in the Arctic, Atmos. Environ., 38, 1153−1166, https://doi.org/10.1016/j.atmosenv.2003.11.016, 2004.

Chung, M. Y., Muthana, S., Paluyo, R. N., and Hasson, A. S.: Measurements of effective Henry's law constants for hydrogen peroxide in concentrated salt solutions, Atmos. Environ., 39, 2981−2989, https://doi.org/10.1016/j.atmosenv.2005.01.025, 2005.

Dul'neva, L. V. and Moskvin, A. V.: Kinetics of formation of peroxyacetic acid, Russ. J. Gen. Chem., 75, 1125−1130, https://doi.org/10.1007/s11176-005-0378-8, 2005.

Kuang, Y., Tao, J. C., Xu, W. Y., Yu, Y. L., Zhao, G., Shen, C. Y., Bian, Y. X., and Zhao, C. S.: Calculating

ambient aerosol surface area concentrations using aerosol light scattering enhancement measurements, Atmos. Environ., 216, 116919, https://doi.org/10.1016/j.atmosenv.2019.116919, 2019.

Li, H., Chen, Z. M., Huang, L. B., and Huang, D.: Organic peroxides' gas-particle partitioning and rapid heterogeneous decomposition on secondary organic aerosol, Atmos. Chem. Phys., 16, 1837−1848, https://doi.org/10.5194/acp-16-1837-2016, 2016.

Li, K., Jacob, D. J., Liao, H., Zhu, J., Shah, V., Shen, L., Bates, K. H., Zhang, Q., and Zhai, S. X.: A two-pollutant strategy for improving ozone and particulate air quality in China, Nat. Geosci., 12, 906−910, https://doi.org/10.1038/s41561-019-0464-x, 2019.

Ma, X. F., Tan, Z. F., Lu, K. D., Yang, X. P., Liu, Y. H., Li, S. L., Li, X., Chen, S. Y., Novelli, A., Cho, C. M., Zeng, L. M., Wahner, A., and Zhang, Y. H.: Winter photochemistry in Beijing: observation and model simulation of OH and $HO_2$ radicals at an urban site, Sci. Total Environ., 685, 85−95, https://doi.org/10.1016/j.scitotenv.2019.05.329, 2019.

Sander, S. P., Abbatt, J., Barker, J. R., Burkholder, J. B., Friedl, R. R., Golden, D. M., Huie, R. E., Kolb, C. E., Kurylo, M. J., Moortgat, G. K., Orkin, V. L., and Wine, P. H.: Chemical kinetics and photochemical data for use in atmospheric studies, Evaluation No. 17, JPL Publication 10-6, Jet Propulsion Laboratory, Pasadena, http://jpldataeval.jpl.nasa.gov (last access: 19 March 2020), 2011.

Sun, X. Y., Zhao, X. B., Du, W., and Liu, D. H.: Kinetics of formic acid-autocatalyzed preparation of performic acid in aqueous phase, Chin. J. Chem. Eng., 19, 964−971, https://doi.org/10.1016/S1004-9541(11)60078-5, 2011.

Wu, Q. Q., Huang, L. B., Liang, H., Zhao, Y., Huang, D., and Chen, Z. M.: Heterogeneous reaction of peroxyacetic acid and hydrogen peroxide on ambient aerosol particles under dry and humid conditions: kinetics, mechanism and implications, Atmos. Chem. Phys., 15, 6851−6866, https://doi.org/10.5194/acp-15-6851-2015, 2015.

Zhou, X. L., Davis, A. J., Kieber, D. J., Keene, W. C., Maben, J. R., Maring, H., Dahl, E. E., Izaguirre, M. A., Sander, R., and Smoydzyn, L.: Photochemical production of hydroxyl radical and hydroperoxides in water extracts of nascent marine aerosols produced by bursting bubbles from Sargasso seawater, Geophys. Res. Lett., 35, L20803, https://doi.org/10.1029/2008GL035418, 2008.

Zhou, X. L., and Lee, Y. N.: Aqueous solubility and reaction kinetics of hydroxymethyl hydroperoxide, J. Phys. Chem., 96, 265−272, https://doi.org/10.1021/j100180a051, 1992.

**Table 3: Calculating the theoretical heterogeneous uptake rate of H₂O₂ on aerosols ($d[X]_p^{t,h}/dt$)[a].**

| Parameters | $T_W$ | $RH$ | $\gamma$ | $S_{aw}$ | $[X]_g$ | $d[X]_p^{t,h}/dt$ |
|---|---|---|---|---|---|---|
| | (K) | (%) | −[b] | (cm²)[c] | (molecules m⁻³)[d] | (ng μg⁻¹ h⁻¹) |
| Averages | 270 | 17.89 | $1.54 \times 10^{-4}$ | 46 | $6.54 \times 10^{14}$ | 0.02 |

[a] These parameters are calculated based on Wu et al. (2015).

[b] $\gamma$ is the heterogeneous uptake coefficient, dimensionless.

[c] $S_{aw}$ is the surface area of aerosols, quoted from Kuang et al. (2019).

[d] $[X]_g$ is the concentration of gas-phase H₂O₂.

**Table 4: Comparison of the H$_2$O$_2$ evolution parameters in the extracted solution among the three types.**

| Parameters | First type | Second type | Third type |
|---|---|---|---|
| Peak time (h) | 5 | 40 | – |
| Decomposition rate of organic peroxides to H$_2$O$_2$ (ng μg$^{-1}$ h$^{-1}$) | 0.01 | 0.10 | – |
| $C_{max}/C_0$ of H$_2$O$_2$ (μM/μM) | 1.52 | 39.22 | 1.00 |
| TPOs/H$_2$O$_2$ (μM/μM) | 5.25 | 40.06 | 47.59 |
| Ratio of decomposable organic peroxides (%) | 29 | 98 | 0 |

---

## Author Response (AR1)

March 20, 2019

ACP Editor

Dear Prof. Nga Lee Ng,

Enclosed please find our revised manuscript entitled "***Partitioning of hydrogen peroxide in gas-liquid and gas-aerosol phases***", revised supplement and two responses to the anonymous referees #1 and #2, respectively. We gratefully thank the reviewers for their constructive suggestions that help us to improve the manuscript. Detailed, point-by-point responses to the comments and corresponding revisions to the manuscript and supplement have been submitted. We sincerely hope that this revised manuscript will be finally acceptable to be published on ACP.

**The major revisions are specified as follows:**

1. We have revaluated the sources and sinks of aerosol-phase $H_2O_2$ based on the formation and consumption rates according to the reviewers' suggestions, and adjusted Table 3 and 4 in the revised manuscript. The ratios of contributions from different pathways have also been modified accordingly.

2. We have calculated the effective Henry's law constant from 2 Jan to 3 Jan 2019 and the contribution of heterogeneous uptake of $HO_2$ to the aerosol-phase $H_2O_2$.

3. We have discussed the influence of the "salting-in" effect on the aerosol-phase $H_2O_2$.

4. We have added some descriptions to prove the validity of the current technique of studying the gas-aerosol partitioning of $H_2O_2$.

5. We have added the experimental conditions of the extracted solution and proved that the derived rates of decomposition can be applied to the ambient atmosphere.

6. We have clarified some numbers (e.g., 88%, 86% and 0.5%), the collection efficiency of organic peroxides, and gas-phase $H_2O_2$ concentration used to estimate the liquid-phase $H_2O_2$.

7. We have explained the reasons for using $8.4 \times 10^4$ M $atm^{-1}$ as Henry's law constant, using the $PM_{2.5}$ concentration as an indicator to calculate the effective gas-aerosol partitioning, the inverse relationship between $H_2O_2$ level and $PM_{2.5}$/sulfate levels, and whether rain samples follow Henry's law.

8. We have corrected the language and grammar errors of the previous manuscript.

9. We have resized Figures 1−6.

**The data corrections are specified as follows:**

1. The aerosol surface area and the gas-phase $H_2O_2$ level in the previous manuscript were calculated incorrectly. We have revised them in Table 3 and revaluated the contribution of heterogeneous uptake of $H_2O_2$ to the aerosol-phase $H_2O_2$.

Detailed changes made in the revised manuscript can be seen in the marked-up version in this response.

Thanks for your time.

Sincerely yours,

Zhongming Chen and co-authors

**Response to Reviewer #1**

We gratefully thank you for your constructive comments and thorough review. Our point-by-point responses can be found below.

(Q=Question, A=Answer, C=Change in the revised manuscript)

**General comments:**

Q1: Xuan et al. performed field measurements of the gas-, liquid- and aerosol-phase $H_2O_2$ in the urban atmosphere of Beijing to understand the partitioning of $H_2O_2$ between gas- and liquid-phase or aerosol-phase. They show that the partitioning of $H_2O_2$ in the gas-liquid phase can be explained by Henry's law and the residual $H_2O_2$ in the raindrops while the aerosol-phase $H_2O_2$ level is significantly higher than that predicted value based on Pankow's absorptive partitioning theory. This paper has important implications for understanding the $H_2O_2$ chemistry and sulfate formation in the atmosphere, so it is well within the scope of ACP. This paper is of great interest to the atmospheric community although some clarifications regarding the data analysis are required. I recommend this paper to be published after addressing the specific comments below.

A1: We highly appreciate your comments and suggestions. The questions you mentioned are specifically answered as follows.

**Specific comments:**

Q2. Estimation of effective partitioning coefficients: The authors determined the gas-aerosol portioning coefficient instead of the effective Henry's law constant for the gas-aerosol phase. Is this due to that aerosol water content cannot be accurately estimated for low RH? The effective Henry's law constant should be estimated for the high RH condition, e.g. heavy haze episodes from 2 Jan to 3 Jan 2019 and compared with the theoretical value.

A2: Yes, you are right. After considering your suggestion, we have calculated the effective Henry's law constant for the gas-aerosol phase during a heavy haze episode from 2 Jan to 3 Jan 2019 and compared it with the theoretical value in the revised manuscript.

C2: Lines 267−272 in Sec. 3.2.1:

"Because aerosol water content (AWC) cannot be correctly evaluated at low RH, the effective field-derived Henry's law constant ($H_P^m$) of $H_2O_2$ was estimated for high RH condition, e.g. a heavy haze episode from 2 January to 3 January 2019 (RH, 30 %). Details regarding the estimation of AWC was shown in the Supplement. It was calculated that AWC, $C_p^m$ and $C_g^m$ levels averaged 3.20 μg m$^{-3}$, 6.63 × 10$^3$ μM, and 1.90 × 10$^{-11}$ atm. Based on Eq. (5), the average $H_P^m$ on 2−3 January 2019 was calculated to be 2.7 × 10$^8$ ± 1.8 × 10$^8$ M atm$^{-1}$. However, the theoretical Henry's law constant ($H_P^t$) at 270 K was 1.1 × 10$^6$ M atm$^{-1}$ (Sander et al., 2011), which was lower than $H_P^m$ by two orders of magnitude."

Q3. The authors estimated that heterogeneous uptake of $H_2O_2$ could account for 86% of the measured $H_2O_2$ in the aerosol phase in Sec 3.2.3 while stated that the heterogeneous uptake of $H_2O_2$ on aerosols contributed less than 0.5% of the aerosol-phase $H_2O_2$ in Sec 3.3. Please clarify.

A3: Thanks for your advice. The two percentages are calculated in different methods. 86 % refers to the ratio of the amount of heterogeneous uptake of $H_2O_2$ to the measured aerosol-phase $H_2O_2$ level, while 0.5 % refers to the ratio of the amount of heterogeneous uptake of $H_2O_2$ to the consumption amount of aerosol-phase $H_2O_2$. In addition, we have revaluated the contribution of the heterogeneous uptake to the aerosol-phase $H_2O_2$ based on the formation and consumption rates according to the reviewers' suggestions, and the heterogeneous uptake could account for 2 % of the consumption rate of the aerosol-phase $H_2O_2$. To avoid confusion, we have removed 86 % in Sec. 3.2.3 and 0.5 % in Sec. 3.3.

Q4. The authors stated that the rates of the decomposition/hydrolysis of organic peroxides in the first and second types were 0.14 ng $\mu g^{-1}$ and 3.65 ng $\mu g^{-1}$ (lines 296−297) and further estimated the contribution of decomposition/hydrolysis of organic peroxides to aerosol $H_2O_2$ to be 32% (lines 343−346). However, these numbers seem to be the steady-state or maximum amount of $H_2O_2$, not formation rates. The estimation should be based on the formation and consumption rate of $H_2O_2$.

A4: Thanks for your suggestion. We have recalculated the estimation considering the formation and consumption rates of $H_2O_2$ and removed the calculation based on the steady-state or maximum amount of $H_2O_2$ in the revised manuscript. Furthermore, we have changed the relevant data in Table 3 and 4.

C4: Lines 417−430 in Sec. 3.3:

"We estimated the contribution of different sources to the aerosol-phase $H_2O_2$ based on the formation and consumption rates. According to the previous estimation of the theoretical sulfate formation rate from January 2 to January 3 2019 (0.29 $\mu g$ $m^{-3}$ $h^{-1}$) and the average mass concentration of $PM_{2.5}$ (106.19 $\mu g$ $m^{-3}$), the consumption rate of $H_2O_2$ should be 0.97 ng $\mu g^{-1}$ $h^{-1}$. With respect to the sources of the aerosol-phase $H_2O_2$, the decomposition/hydrolysis of organic peroxides was firstly considered, with average rates of the rising stage for the first and second types (Fig. 6), 0.01 ng $\mu g^{-1}$ $h^{-1}$ and 0.10 ng $\mu g^{-1}$ $h^{-1}$, respectively. Because the extracted solution was stored under 255 K, lower than the actual atmospheric temperature (270 K), the decomposition/hydrolysis rates of organic peroxides were underestimated and an adjusting factor should be multiplied. The factors for the three typical labile organic peroxides (HMHP, PFA, and PAA) were 13, 3, and 2, respectively, as shown in the Supplement. Assuming the factor was in the range of 2−13, the average decomposition/hydrolysis rate of organic peroxides for the first and second types (0.055 ng $\mu g^{-1}$ $h^{-1}$) was used to calculate the formation rate. The formation rate of the aerosol-phase $H_2O_2$ from the decomposition/hydrolysis of organic peroxides could account for 11−74 % of the consumption rate by sulfate formation. Moreover, the heterogenous uptake of $HO_2$ and $H_2O_2$ were also likely to improve the aerosol-phase $H_2O_2$ level at the rates of 0.22 ng $\mu g^{-1}$ $h^{-1}$ and 0.02 ng $\mu g^{-1}$ $h^{-1}$, respectively, which can offset 22 % and 2 % of the consumption rate of $H_2O_2$, respectively."

Q5. Though the heterogeneous uptake of $HO_2$ on aerosols is not well understood, it is possible to estimate its contribution to aerosol $H_2O_2$ using the reactive uptake coefficient of $HO_2$ to aerosol from literature and assuming the product to be $H_2O_2$ (Li et al., 2019). It is recommended to perform such calculations to provide more insights.

A5: Thanks for your advice. We have added the calculation of the heterogeneous uptake of $HO_2$ on

aerosols in the revised manuscript.

C5: Lines 325−331 in Sec. 3.2.3:

"As $HO_2$ radical is a precursor of $H_2O_2$, the heterogeneous uptake of $HO_2$ onto aerosols may also contribute to the formation of the aerosol-phase $H_2O_2$. We assumed that the reactive uptake coefficient of $HO_2$ to aerosol particles was 0.2, and the product of $HO_2$ was $H_2O_2$ (Li et al., 2019). At the same observation site in winter of 2017, $HO_2$ concentration for noontime averaged $(0.4 \pm 0.2) \times 10^8$ $cm^{-3}$ and $(0.3 \pm 0.2) \times 10^8$ $cm^{-3}$ on clean and polluted days, respectively (Ma et al., 2019). Since $HO_2$ level data in 2018 was not available, we used the level of $HO_2$ on clean days in winter of 2017 for calculations, and the average was about $0.2 \times 10^8$ $cm^{-3}$ at day-time. The heterogenous uptake rate of $HO_2$ on aerosols was calculated the same way as $H_2O_2$, and the formation rate of the aerosol-phase $H_2O_2$ by reactive uptake of $HO_2$ averaged 0.22 ng $\mu g^{-1}$ $h^{-1}$ at all day."

Q6. The authors should discuss the "salting in" effect of high ionic strength of aerosol particles on gas-aerosol partitioning of $H_2O_2$ though it may only have a minor contribution to the enhanced aerosol $H_2O_2$ concentrations.

A6: Thanks for your suggestion. We have discussed the "salting-in" effect of high ionic strength of aerosol particles on the gas-aerosol partitioning of $H_2O_2$ in the revised manuscript.

C6: Lines 272−278 in Sec. 3.2.1:

"In Chung's study (2005), "salting-in" effect can improve the level of $H_2O_2$ by a factor of two when the concentrations in salt solutions were up to 10 M, and the most obvious "salting-in" effect of salt solutions was ammonium sulfate. In this paper, the levels of aerosol-phase $NH_4^+$ and $SO_4^{2-}$ on 2−3 January 2019 were 94 M and 21 M, respectively, and the level of $(NH_4)_2SO_4$ was assumed to be 21 M. The increasement of $H_P^m$ by the "salting-in" effect of $(NH_4)_2SO_4$ was about $3.2 \times 10^6$ M $atm^{-1}$ at 286 K based on equations in Chung et al. (2005). Even though aerosol particles were collected at $270 \pm 4$ K and the increasement may be greater, the "salting-in" effect could not fully explain the difference between $H_P^m$ and $H_P^t$. Other sources need to be found later."

Q7. Line 82: Are the organic peroxide concentrations corrected for the collection efficiency?

A7: Thanks for your suggestion. Because the measured concentrations of organic peroxides were near the detection limit of the HPLC method in BJ-2018Winter measurement, the levels of organic peroxides were not discussed in this paper. Alternatively, we used the concept of total peroxides as a measure to estimate the sources of the aerosol-phase $H_2O_2$ in Sec. 3.2. The level of total peroxides was measured using the iodometric spectrophotometric method with an extraction efficiency close to ~ 98 % (Li et al., 2016), so we did not correct the total peroxides level.

Q8. Lines 149−150: Please explain how 88% is derived.

A8: Thanks for your suggestion and we have explained it in line 166 in the revised manuscript. The percentage was calculated as the average ratio of the predicted level to measured level of $H_2O_2$ in each rain sample. The value indicated that 88 % of the liquid-phase $H_2O_2$ in all the rain samples collected was from gas-phase partitioning. Since the measured $H_2O_2$ level in some rain samples was lower than the

predicted value, gas-phase partitioning accounted for a high proportion in all samples based on statistics on averages.

Q9. Line 181: What is the gas-phase $H_2O_2$ concentration used to estimate the liquid-phase $H_2O_2$?.

A9: The gas-phase $H_2O_2$ concentration used to estimate the liquid-phase $H_2O_2$ was $0.30 \pm 0.26$ ppbv, and we have added it in lines 203−204 in the revised manuscript. We assumed that the gas-phase $H_2O_2$ concentration was homogeneous, and the gas-phase $H_2O_2$ in the cloud atmosphere was equal to the gas-phase $H_2O_2$ near the ground (line 191).

Q10. Section 3.2.4: The experimental details on the decomposition of organic peroxides should be provided. Is the extracted solution exposed to light at room temperature? Are these experiments conducted at atmospheric relevant conditions so that the derived rates of decomposition can be applied to ambient?

A10: Thanks for your suggestion. The extracted solution was away from light at 255 K. The experimental conditions were chosen based on certain considerations, which have been added in the revised manuscript and Supplement. The influence of experimental conditions on the derived rates has also been discussed in the revised manuscript and Supplement. We think that the derived rates of decomposition can be applied to the ambient atmosphere.

C10: Lines 111−115 in Sec. 2.2.3:

"The remaining extracted solution was stored at 255 K away from light for subsequent measurement of $H_2O_2$ concentration variation with time, and details of the experimental conditions of the extracted solution are shown in the Supplement. Photochemical reactions of aerosols may produce aerosol-phase $H_2O_2$ (Zhou et al., 2008), and the effect of the photochemical reactions on the level of $H_2O_2$ in the extracted solution was discussed in the Supplement."

Lines 1−48 in the Supplement:

"The extracted solution was stored under refrigeration at 255 K, away from light. The first reason to choose 255 K was that the temperature during BJ-2018Winter measurement averaged 270 K, less than 273 K. The second reason was that under 255 K, the decomposition rate of $H_2O_2$ should be reduced, which contributed to a more accurate estimation of the decomposition rates of organic peroxides to $H_2O_2$. Li et al. (2016) studied the stability of $H_2O_2$ in SOA stored on-filter at 255 K and 298 K. It was found that the level of $H_2O_2$ remained stable for 6 days at 255 K but decreased gradually at 298 K. The third reason was that the $H_2O_2$ level in the extracted solution was very low at time=0, which could easily decompose at 277 K, therefore, the extracted solution should be stored at 255 K.

Due to the positive correlation between temperature and decomposition rates, the derived rates of decomposition in this paper were lower than the actual rates of decomposition. To discuss the influence of the storage temperature on the decomposition rates of organic peroxides, hydroxymethyl hydroperoxide (HMHP), peroxyformic acid (PFA), and peroxyacetic acid (PAA) were chosen as representatives. According to the Arrhenius equation, the reaction rate usually increases exponentially as temperature increases. The ratios of the decomposition/hydrolysis rates of HMHP, PFA, and PAA at 270 K to 255 K were 13, 3, and 2, respectively (Zhou and Lee, 1992; Dul'neva and Moskvin, 2005; Sun et

al., 2011). We have considered the influence of temperature on the decomposition rates of organic peroxides when calculating the aerosol-phase $H_2O_2$ formation rate from the decomposition/hydrolysis of organic peroxides, as shown in Sec. 3.3.

The extracted solution was away from light in this paper, which was different from atmospheric relevant conditions (i.e., exposure to sunlight in the ambient atmosphere), and may affect the data applicability in this study. We chose this experimental condition because if the extracted solution was exposed to sunlight, the photochemical reactions of organic matters and the decomposition/hydrolysis of organic peroxides will coexist, and we cannot distinguish the effects of these two processes. By doing so, the specific contribution of the decomposition/hydrolysis of organic peroxides to the aerosol-phase $H_2O_2$ was estimated. With respect to the photochemical reactions of organic matters, Zhou et al. (2008) have discussed that as the exposure time of the extracted solution to sunlight increased, the production of peroxides in nascent marine aerosols first increased rapidly and then slowly. The change trend in Zhou's study was the same as that of the aerosol-phase $H_2O_2$ level in this paper (Fig. 6). The estimated 24-h-average rate of $H_2O_2$ photochemical production in Alert particles was about 9 mM h$^{-1}$ at 248 K (Anastasio and Jordan, 2004). We assumed that the photoformation rate of $H_2O_2$ in Beijing particles was also 9 mM h$^{-1}$. And the concentrations of AWC and $PM_{2.5}$ from 2 January to 3 January 2019 were 3.20 μg m$^{-3}$ and 90.36 μg m$^{-3}$, respectively. The formation rate of the aerosol-phase $H_2O_2$ from photochemical reactions was estimated to be 0.011 ng μg$^{-1}$ h$^{-1}$ at 248 K. In addition, the activation energy of $H_2O_2$ photoformation was 9 kJ mol$^{-1}$ (Anastasio et al., 1994), and the rate of $H_2O_2$ photoformation at 270 K should be 1.4 times higher than the value at 248 K. Compared with the aerosol-phase $H_2O_2$ formation rates from the decomposition/hydrolysis of organic peroxides and the heterogeneous uptake of $HO_2$, the $H_2O_2$ photoformation could be neglected.

Based on the above analysis, we believe that the derived rates of decomposition under the experimental conditions in this paper can be applied to the ambient atmosphere."

**Technical corrections:**

Q11. Lines 59−60: References are missing.

A11: Thanks for your reminder. We have added related references in line 65.

Q12. Equation 4: TSP or $PM_{2.5}$ should be used instead of $C_{om}$.

A12: Yes, we have changed "$C_{om}$" into "TSP" in Eq. (4) in the revised manuscript, and used the $PM_{2.5}$ mass concentration as an indicator because the $PM_{2.5}$ concentration was available based on our measurements.

Q13. Line 331: "measured" should be "was measured to be".

A13: Thanks for your advice. We have revised it in line 391.

**Table 3: Calculating the theoretical heterogeneous uptake rate of H₂O₂ on aerosols ($d[X]_p^{t,h}/dt$)[a].**

| Parameters | $T_W$ (K) | $RH$ (%) | $\gamma$ $-$[b] | $S_{aw}$ (cm²)[c] | $[X]_g$ (molecules m⁻³)[d] | $d[X]_p^{t,h}/dt$ (ng μg⁻¹ h⁻¹) |
|---|---|---|---|---|---|---|
| Averages | 270 | 17.89 | $1.54 \times 10^{-4}$ | 46 | $6.54 \times 10^{14}$ | 0.02 |

[a] These parameters are calculated based on Wu et al. (2015).
[b] $\gamma$ is the heterogeneous uptake coefficient, dimensionless.
[c] $S_{aw}$ is the surface area of aerosols, quoted from Kuang et al. (2019).
[d] $[X]_g$ is the concentration of gas-phase H₂O₂.

**Table 4: Comparison of the $H_2O_2$ evolution parameters in the extracted solution among the three types.**

| Parameters | First type | Second type | Third type |
|---|---|---|---|
| Peak time (h) | 5 | 40 | − |
| Decomposition rate of organic peroxides to $H_2O_2$ (ng $\mu g^{-1}$ $h^{-1}$) | 0.01 | 0.10 | − |
| $C_{max}/C_0$ of $H_2O_2$ (μM/μM) | 1.52 | 39.22 | 1.00 |
| TPOs/$H_2O_2$ (μM/μM) | 5.25 | 40.06 | 47.59 |
| Ratio of decomposable organic peroxides (%) | 29 | 98 | 0 |

**Response to Reviewer #2**

We gratefully thank you for your constructive comments and thorough review. Our point-by-point responses can be found below.

(Q=Question, A=Answer, C=Change in the revised manuscript)

Q1: This manuscript by Xuan et al describes a detailed study of the partitioning of $H_2O_2$ in the atmosphere through field measurements. The authors quantified $H_2O_2$ in the gas phase, aerosol, and rainwater (as a surrogate for cloud water). By comparing the measured and theoretical Henry's law constant, as well as the measured and theoretical partitioning coefficient, the authors conclude that the measured values for both are higher than the theoretical values. An in-depth assessment is conducted to evaluate the influence of raindrop falling on the quantified $H_2O_2$ concentration, as well, a discussion on the source and sink of $H_2O_2$ in aerosol is provided. $H_2O_2$ plays an important role in the atmosphere, and understanding its partitioning in different atmospheric phases is of great importance for the atmospheric chemistry community. The manuscript is within the scope of ACP. The data analysis and calculation were performed with caution. I recommend publication on ACP after addressing the following comments.

A1: We highly appreciate your comments and suggestions. The questions you mentioned are specifically answered as follows.

**Major comment:**

Q2. In section 3.2.4, the authors present the evolution of $H_2O_2$ as a function of time in the aerosol abstract, and a detailed discussion on the potential source of $H_2O_2$. This result highlights the challenges in making off-line $H_2O_2$ measurement from filter samples. Especially, when the sampling time is as long as 11.5 h (Line 100), it is very likely that the organic peroxides present in the aerosol sample is continuously decomposing on the filter. The authors categorize the $H_2O_2$ evolution into three types and postulate the relevant source of $H_2O_2$ for each type. However, in my opinion, this appears too speculative. The decomposition of $H_2O_2$ on filter is difficult to control, and the quantified $H_2O_2$ could be merely a snapshot of an ongoing decomposition process. The authors must justify whether it is valid at all to establish gas-aerosol partitioning of $H_2O_2$ based on the current technique.

A2: Thanks for your suggestion. We have deleted the relevant source of $H_2O_2$ in each type in Table 4 in the revised manuscript. Because organic peroxides are unstable and can easily decompose, off-line measurement of the aerosol-phase $H_2O_2$ could only obtain a snapshot of the decomposition process. Although there may be uncertainties regarding the aerosol-phase $H_2O_2$ measurement and the calculation of gas-aerosol partitioning coefficient of $H_2O_2$, this paper provides new insights into understanding the gas-aerosol partitioning of $H_2O_2$, as well as the sources and sinks of aerosol-phase $H_2O_2$, which may contribute to the future studies related.

Provided that the influence of Teflon filters on the reactions of aerosol particles is so little as to be unnoticeable, the decomposition/hydrolysis rates of organic peroxides in aerosol particles on the filters are same as that in the atmosphere. It is well known that the decomposition/hydrolysis rates of organic peroxides are often positively related to the levels of organic peroxides. Due to the low aerosol water content of particles, the concentrations of aerosol-phase organic peroxides were ~ 5 orders of magnitude

higher than that in the extracted solution, which were estimated based on a comparison between the amount of the extracted solution and aerosol water content. The actual decomposition/hydrolysis rates in aerosol particles may be higher than that in the extracted solution. However, we cannot know how much the difference between them due to the limitations of the available measurement technique. Based on above analysis, to a large extent, the effective gas-aerosol partitioning coefficient estimated in this paper can represent the actual gas-aerosol partitioning of $H_2O_2$.

Furthermore, it took around 40 min to extract and transport the sample to the observation site for $H_2O_2$ measurement. Organic peroxides in the extracted solution may decompose into $H_2O_2$ during the process, leading to overestimation of the effective gas-aerosol partitioning coefficient of $H_2O_2$. Provided that the maximum decomposition/hydrolysis rate of organic peroxides was 0.10 ng $\mu g^{-1}$ $h^{-1}$ (line 355 in the revised manuscript), the corrected gas-aerosol partitioning coefficient averaged $6.9 \times 10^{-4}$ $m^3$ $\mu g^{-1}$, which was the lowest value due to the assumed maximum value of the decomposition/hydrolysis rate of organic peroxides. Because the corrected value of the effective gas-aerosol partitioning coefficient was still much higher than $K_P^t$, we did not correct the data. The above analysis has been added in the Supplement (lines 158−168).

In addition, the level of gas-phase $H_2O_2$ during BJ-2018Winter was very low, only tens of pptv. Lengthening sampling time will increase the aerosol-phase $H_2O_2$ concentration and ensure accurate quantitative detection of $H_2O_2$, but it will also introduce some unknown errors. Therefore, we will comprehensively consider to determine an optimal sampling time in the future study of the gas-aerosol partitioning of $H_2O_2$.

**Minor comments:**

Q3. Literature-reported Henry's law constants of $H_2O_2$ varies across a certain range. The authors should justify why they used $8.4 \times 10^4$ M/atm. Is this the recommended value by the JPL publication?

A3: Thanks for your suggestion. Henry's law constant of $H_2O_2$ ($8.4 \times 10^4$ M $atm^{-1}$) used in this paper was quoted from Sander et al. (2011), which was published in JPL Publication 10-6. In addition, Sander (2015) sorted Henry's law constants of $H_2O_2$ based on the data reliability, and $8.4 \times 10^4$ M $atm^{-1}$ ranked higher. In addition, the latest recommended value was $8.7 \times 10^4$ M $atm^{-1}$ at 298 K (Burkholder et al., 2015), which was close to $8.4 \times 10^4$ M $atm^{-1}$ used in this paper.

Q4. Line 139 – Is the $PM_{2.5}$ concentration a good indicator for $C_{om}$ in a polluted environment like Beijing?

A4: Thanks for your suggestion. In previous studies, they used TSP in calculating the field-derived gas-aerosol partitioning coefficient and assumed that the weight fraction of the organic matter phase in TSP was 1 (Pankow et al., 1994; Odum et al., 1996; Shen et al., 2018; Qian et al., 2019). We have replaced "$C_{om}$" with "TSP" in Eq. (4) in the revised manuscript, and used the $PM_{2.5}$ mass concentration since TSP concentration was not available.

Q5. Line 150 – The authors state that when $H_A^m$ was less than $H_A^t$, the samples followed Henry's law. Why? Shouldn't they agree (neither higher nor lower)?

A5: Thanks for your suggestion. The previous expression was inappropriate and we have redefined

whether rain samples followed Henry's law in the revised manuscript.

C5: Lines 168−171 in Sec. 3.1.1:

"We divided 52 rain samples into three types based on the comparison of the measured and predicted levels of $H_2O_2$. When the difference between levels of the measured and predicted liquid-phase $H_2O_2$ fell within ± 20 %, we suggested that these samples (Type B) followed Henry's law, and the remaining samples (Type A and C) did not agree with Henry's law. The percentages of samples in Type A, B and C were 69 %, 19 %, and 12 %, respectively."

Q6. Line 275 – The authors report here that heterogeneous uptake can count for 86% of aerosol phase $H_2O_2$. Later in Line 346, the author report 0.5%. Please clarify.

A6: Thanks for your suggestion. The two percentages are calculated in different methods. 86 % refers to the ratio of the amount of heterogeneous uptake of $H_2O_2$ to the measured aerosol-phase $H_2O_2$ level, while 0.5 % refers to the ratio of the amount of heterogeneous uptake of $H_2O_2$ to the consumption amount of aerosol-phase $H_2O_2$. In addition, we have revaluated the contribution of the heterogeneous uptake to the aerosol-phase $H_2O_2$ based on the formation and consumption rates according to the reviewers' suggestions, and the heterogeneous uptake could account for 2 % of the consumption rate of the aerosol-phase $H_2O_2$. To avoid confusion, we have removed 86 % in Sec. 3.2.3 and 0.5 % in Sec. 3.3.

Q7. Line 370 – correct me if I am wrong. "additional source of liquid-phase $H_2O_2$ gradually increased" – should this be the sink of $H_2O_2$ due to droplet-to-gas transfer is gradually reduced?

A7: Yes, you are right. We have rewritten the description in the revised manuscript.

C7: Lines 450−451 in Sec. 4:

"In addition, the sink of $H_2O_2$ due to droplet-to-gas transfer was reduced with an increase in raindrop diameter, thus the liquid-phase $H_2O_2$ level also increased."

Q8. Figure 5, and Line 263 – the authors interpret the inversely related $H_2O_2$ concentration and $PM_{2.5}$/sulfate concentrations as results of a $H_2O_2$ sink by $SO_2$ oxidation. However, could the inversed relation be just due to dilution of $H_2O_2$ when aerosol loading is high?

A8: Thanks for your suggestion. We did not consider the dilution of $H_2O_2$ due to an increase in aerosol loading, and we have added it in the revised manuscript. To avoid the effects of outliers, we chose $10^{th}$ and $90^{th}$ percentiles of the levels of aerosol-phase $H_2O_2$, $SO_4^{2-}$ and $PM_{2.5}$ to explain the inverse relationship between $H_2O_2$ and $SO_4^{2-}$/$PM_{2.5}$. The extent of the concentration variations of $H_2O_2$, $SO_4^{2-}$ and $PM_{2.5}$ were 22, 6 and 5, respectively. Because the level of $H_2O_2$ changed more than that of $SO_4^{2-}$ and $PM_{2.5}$, the inverse relationship still existed when we eliminated the interference of the dilution effect due to the high aerosol loading. In addition, the ratios of the extent of the concentration variations between $H_2O_2$ and $SO_4^{2-}$/$PM_{2.5}$ were equal to 4, indicating that a $H_2O_2$ sink by $SO_2$ oxidation was more important than the dilution effect and the dilution effect could be neglected.

C8: Lines 303−306 in Sec. 3.2.2:

"The extent of the concentration variations of $H_2O_2$, $SO_4^{2-}$ and $PM_{2.5}$ at $10^{th}$ and $90^{th}$ percentiles were 22, 6 and 5, respectively, suggesting that the inverse relationship still existed when we eliminated the

interference of the dilution effect due to a high aerosol loading. The dilution effect was unimportant and could be neglected."

**Technical comments:**

Q9. Line 59 – "easily to absorbed" to "easily absorbed".

A9: Yes, we have revised it in line 65.

Q10. Line 144 – "statistically counted" appears awkward. Should probably remove.

A10: Yes, we have removed it in line 160.

Q11. Line 146 – "with" 25.20 $\mu$M – is 25.20 $\mu$M the theoretical value? "with" makes the sentence unclear.

A11: Yes, we have replaced "with" with "is" in line 162.

Q12. Line 171 – "almost" less than 2000 m – should this be "always" less than 2000 m?

A12: Yes, we have changed "almost" into "always" in line 194.

[revised manuscript text omitted]